# Efficient flow equations for dissipative systems

**Gary Schmiedinghoff⋆ and Götz S. Uhrig**

Condensed Matter Theory, TU Dortmund University,
Otto-Hahn-Straße 4, 44227 Dortmund, Germany

⋆ gary.schmiedinghoff@tu-dortmund.de

## Abstract

Open quantum systems provide an essential theoretical basis for the development of novel quantum technologies, since any real quantum system inevitably interacts with its environment. Lindblad master equations capture the effect of Markovian environments. Closed quantum systems can be treated using flow equations with the particle conserving generator. We generalize this generator to non-Hermitian matrices and open quantum systems governed by Lindbladians, comparing our results with recently proposed generators by Rosso et al. [1]. In comparison, we find that our advocated generator provides an efficient flow with good accuracy in spite of truncations.

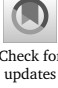
# 1 Introduction

Even with recent computational advances, the theoretical study of many-body quantum physics proves computationally very challenging, in particular for open systems. Often, only part of the system is modeled microscopically with the remaining system treated as an external environment, introducing dissipation to the microscopic system. The environment cannot be

disregarded, since in Nature as well as in practical technical applications one can never completely isolate a system from its surroundings, a problem especially significant for quantum information processing [2].

Lindblad (or Gorini-Kossakowski-Sudarshan-Lindblad) master equations provide the most general description of systems with couplings to Markovian baths [3]. The corresponding Lindblad operator is non-Hermitian, as opposed to the Hermitian Hamiltonian and observables used in closed quantum mechanical models. Prominent examples of such systems include exciton polaritons [4], cold gases with losses [5], circuit QED array [6], trapped ions [7] and Rydberg atoms [8]. Outside of Lindblad formalism, non-Hermitian Hamiltonians can appear systematically [9, 10] as well , for instance in the Dyson-Maleev representation, in which the spin operators are replaced by effective bosonic operators which are not explicitly Hermitian on the full bosonic Hilbert space [11, 12]. To solve the dynamics of such systems, novel methods must be developed or existing methods for treating Hermitian Hamiltonians must be adapted. Various methods have been introduced to this end, including quantum trajectories [13], tensor networks [14, 15] and extensions of mean field theories [16, 17]. Efficient and robust methods that can easily be extended to more classes of open quantum system are invaluable in understanding novel quantum phenomena, especially for systems which cannot be solved exactly and need to be treated approximately.

The method of interest in this paper is the *flow equation* approach [18], also known as *continuous unitary transformations* (CUTs), a renormalization scheme which is traditionally used to transform Hermitian matrices and operators into an effective basis. Flow equations were proposed by Wegner for condensed matter physics [19] and independently by Glazek and Wilson for high-energy physics under the name *similarity renormalization scheme* [20, 21]. Similar flow equations have been studied by the mathematicians Brockett, Chu und Driessel, called *double bracket flow* [22–24]. Flow equations have been applied on a wide range of problems including the Anderson model [25, 26], the spin-boson model [27], electron-phonon interaction [28, 29], quantum systems including an environment [30, 31], spin chains with and without frustration [32–35], the quantum sine-Gordon models [36, 37], Shastry-Sutherland lattices [38], spin ladders in copper nitrate [39], coupled spin ladders in the compound $BiCu_2PO_6$ [40], the Kondo model out of equilibrium [41], and a quenched Hubbard model [42]. Very often the goal is to decouple subspaces $M_n$ of different numbers $n$ of quasi-particles by the change of basis. The transformation to the decoupling basis for the Hamiltonian has to be used for observables as well [30, 43]. In the quasi-particle conserving basis, subsequent calculations such as the calculation of spectral functions can be performed computationally efficiently and with high numerical accuracy [44] and the effects from various quasi-particle spaces can be switched on or off selectively [45].

The dynamics of extended physical systems are described by large matrices with exponentially increasing dimension for increasing system size. Such matrices cannot be dealt with numerically. Hence, physically justifiable truncation schemes are required to obtain closed sets of equations. This applies equally to descriptions in second quantization where one has to truncate the tracked number of operators to a finite set of operator monomials. Truncations can be performed based on the range of processes in real-space [46], perturbation theory [32] or scaling arguments such as the operator product expansion [36, 37] or the scaling dimension [47, 48]. Flow equations are especially useful for systems that can be expanded perturbatively [49]. By using generator schemes with renormalizing properties and preserving band-diagonality, the error introduced by truncating the system to only the most relevant components can be kept small [50]. The flow equation approach can also be extended and combined with other formalisms such as the Floquet theory [51–53].

So far, flow equations are restricted to Hermitian matrices or non-Hermitian matrices with real eigenvalues. But in dissipative open quantum systems one encounters non-Hermitian

matrices with complex eigenvalues. The imaginary parts encode relaxation and dissipation. Their quantitative understanding is crucial in non-equilibrium physics in general, for instance for any pump-probe experiments, and for quantum coherent control as it matters for quantum information processing. For this reason, we want to generalize the framework of flow equations to *dissipative flow equations* [1] which work for non-Hermitian matrices as well. Dissipative flow equations can be used for studying the non-unitary dynamics of a dissipative system. This is different from approaches in the past, which attempted to describe the unitary dynamics of system and environment microscopically as a whole [26,27,30,54–62]. The idea of dissipative flow equations is to start from a framework of open quantum systems and to solve the problem using a generalized flow equation scheme. This field is still in its infancy and this paper aims to advance it by advocating an improved generator.

In 2020, Rosso et al. laid an important foundation for dissipative flow equations by introducing three novel generators for non-Hermitian matrices [1]. The authors focused on the important speed of convergence of the flows induced by the generators for various systems. In the present work, we extend this focus to the equally crucial accuracy of the flows in spite of truncations. As exposed above, truncations are inevitable for almost all applications. Thus, the robustness of the flows against truncation is very important; an appropriate generator scheme must keep the truncation error small to capture the physics correctly. The key idea to achieve this consists in the renormalization properties of the generator: large energies and rates are dealt with before the flow simplifies the processes at low energies and rates. To this end, we propose an improved generator scheme, the *generalized particle conserving* generator (gpc-generator), which addresses both the requirements for convergence speed and accuracy. We benchmark both criteria for all four generators (the three from Rosso et al. and the gpc-generator) by applying them to various mathematical and physical systems. Our results show that the previously proposed generators only fulfill one of the two criteria, respectively, while the gpc-generator allows for a good tradeoff, combining an efficient numerical calculation with low truncation errors.

We begin by introducing the basic flow equation mechanism in Sec. 2.1. We then recall the *particle conserving generator* (pc-generator) [32,50] in Sec. 2.2. The pc-generator is known for an improved energy-dependence of the convergence compared to the Wegner generator [19] and for preserving the band-diagonality of the Hamiltonian. Then, we introduce the basic concept of non-Hermitian flow equations in Sec. 2.3 and briefly discuss the important physical application of the Lindblad master equations in Sec. 2.4. We show how the pc-generator fails for such non-Hermitian matrices in Sec. 2.5 and point out the solution for the trivial special case of Antihermitian matrices in Sec. 2.6. We conclude the discussion of the pc-generator by explaining how perturbative truncations can be performed in Sec. 2.7.

To tackle the non-Hermitian case, we introduce the *generalized particle conserving generator* (gpc-generator) in Sec. 3.1. We proceed by comparing the gpc-generator with the generators proposed by Rosso et al. [1], which are generalizations of the generator introduced by Wegner [19] and the one by White [63] to non-Hermitian matrices, in Sec. 3.2. We prove convergence of the gpc-generator perturbatively in Sec. 3.3.

In Sec. 4, we benchmark and compare the four generators for concrete mathematical and physical models. We start by introducing the benchmark parameters in Sec. 4.1 and by discussing a simple physical two-level model analytically in Sec. 4.2. We proceed with a numerical benchmark of a mathematical example of randomly sampled matrices in Sec. 4.3, where we also show the importance of the ordering of the diagonal matrix elements. We proceed with the numerical benchmark for two physical examples, both exhibiting dissipation and a single strongly dissipative state: One is a model with ordered energies in Sec. 4.4 and the other is a model with unordered energies in Sec. 4.5. As a final numerical example we sample random Lindbladians in Sec. 4.6. We summarize our findings in Sec. 5.

## 2 Fundamental Challenges

### 2.1 Hermitian Flow Equations

The idea of the flow equation method or continuous unitary transformations is to transform a problem described by one or multiple Hermitian matrices to a basis, in which the matrix takes a simpler and more tractable form. In that sense, the key idea is to transform a Hermitian matrix $H$ to an effective matrix $H_{\text{eff}}$ using unitary transformations. In parallel, all relevant observables $O$ are transformed to effective observables $O_{\text{eff}}$ using the same basis as $H_{\text{eff}}$. Instead of doing this in a single step, we perform a continuous transformation [19, 20, 32, 50].

$$H(\ell) = U(\ell)H(0)U(\ell)^{\dagger}, \qquad H(0) := H, \quad H(\infty) =: H_{\text{eff}}, \tag{1a}$$

$$U(\ell) := \mathcal{T}_{\ell} \exp\left(\int_0^{\ell} \eta(\ell')d\ell'\right), \tag{1b}$$

with an unitary transformation matrix $U(\ell)$ and the $\ell$-ordered product $\mathcal{T}_{\ell}$. Note that $U(\ell)$ can be written in terms of the generator $\eta(\ell)$ as in (1b). While this approach seems to introduce unnecessary complexity to the problem, it allows us to write the problem in the form of the flow equations

$$\frac{\mathrm{d}}{\mathrm{d}\ell}H(\ell) = \left[\eta[H(\ell)], H(\ell)\right], \tag{2}$$

which can be solved analytically in some cases, but is most commonly solved numerically. Here, we use the notation $\eta[H(\ell)]$ to emphasize that the generator generally depends on $H(\ell)$. Analogously, the flow equations of the observables read

$$\frac{\mathrm{d}}{\mathrm{d}\ell}O(\ell) = \left[\eta[H(\ell)], O(\ell)\right]. \tag{3}$$

The advantage of an $\ell$-dependent generator can be illustrated by the analogy to a rotation. The generator represents the rotation axis. Its dependence on $\ell$ allows one to optimize the orientation of rotation to the momentary situation at this particular $\ell$. In the extremely high-dimensional space of operators this optimization represents a vital asset. Thus, by choosing an optimized generator scheme, we modify the properties of the flow in order to obtain an amenable effective matrix $H_{\text{eff}}$. For instance, the matrix $H$ can be the Hamilton operator of a many-particle system which does not conserve the number of particles, while a better tractable effective matrix $H_{\text{eff}}$ conserves the number of quasi-particles and allows for a much simpler subsequent analysis.

### 2.2 The PC-Generator

The particle conserving generator, in the most general form, is defined by

$$\eta^{\text{pc}}[H] = H^+ - H^-, \tag{4a}$$

$$\eta^{\text{pc}}_{nm}[H] = \text{sign}(q_{nn} - q_{mm})h_{nm}, \tag{4b}$$

for Hermitian matrices $H$ [32, 50, 64]. We denote the matrix elements of $\eta$ by $\eta_{nj}$. The dependencies on $\ell$ are not denoted explicitly for brevity. The concrete definition of the operator $Q$ is chosen according to the specific problem, see below. The operator $Q$ is diagonal; it is not transformed together with $H$, which means that $Q$ stays diagonal during the flow. The factors $q_{nn}$ are the diagonal elements of $Q$ and the matrix $H^+$ contains all terms of $H$ which increase

$Q$, while $H^-$ contains all terms which decrease $Q$. In matrix representation, the elements $h^{\pm}_{nm}$ of $H^{\pm}$ are

$$h^{\pm}_{nm} = \begin{cases} h_{nm} & \forall \mathrm{sign}(q_{nn} - q_{mm}) = \pm 1\,, \\ 0 & \text{else}\,. \end{cases} \tag{5}$$

We can choose the concrete definition of operator $Q$ depending on the desired $H_{\mathrm{eff}}$. For example, we can have $Q$ count the number of quasi-particles if we want to decouple subspaces with different numbers of quasi-particles, obtaining a quasi-particle conserving $H_{\mathrm{eff}}$. In that case, $H^+$ contains all terms which increase the number of quasi-particles and $H^-$ all terms which reduce the number of quasi-particles. If we want to diagonalize $H$, we choose $q_{nn} := n$ and obtain

$$\boxed{\eta^{\mathrm{pc}}_{nj}[H] = \mathrm{sign}(n - j) h_{nj}\,,} \tag{6}$$

in which case $H^+$ and $H^-$ are the upper and lower triangular part of $H$, respectively. We will use definition (6) of the pc-generator in this work, as it is directly accessible for numerical benchmarks on various systems. We stress that because of the term $\mathrm{sign}(n - j)$, the generator attempts to sort the diagonal elements in ascending order [32, 50, 65]. If they are not sorted initially or the order is destroyed because two diagonal elements swap indices in the course of the flow, the flow performs major rearrangements of matrix elements. Temporarily, this causes an increase of off-diagonal elements. One possibility to avoid this consists in defining $q_{nn} = h_{nn}$, yielding

$$\eta^{\mathrm{pc}}_{nm}[H] = \mathrm{sign}(h_{nn} - h_{mm}) h_{nm}\,. \tag{7}$$

Note that this does not offer a fundamental improvement of the method and works in the same way as interchanging the basis vectors of the states without sorted $h_{nn}$.

In comparison to the Wegner generator [19]

$$\hat{\eta}_{\mathrm{W}}[\hat{H}] = [\hat{H}_{\mathrm{diag}}, \hat{H}_{\mathrm{non\text{-}diag}}]\,, \tag{8}$$

the pc-generator (6) conserves the band-diagonality of $H$. When the flow is close to a fixed point, off-diagonal elements converge to 0 according to $m_{nj}(\ell) \propto \exp(-|\Delta E_{nj}|\ell)$ with diagonal differences $\Delta E_{nj} := m_{nn} - m_{jj}$ as opposed to the $m_{nj} \propto \exp(-|\Delta E_{nj}|^2 \ell)$ scaling of the Wegner approach. Diagonalization can be achieved with the pc-generator even when $H$ features degeneracies [32,50,65]. Since the pc-generator was first suggested, several perturbative modifications [66] and other improvements have been established, for instance, the directly evaluated enhanced perturbative CUT (deepCUT) [49].

For Hermitian matrices $H$, the pc-generator induces the flow

$$\partial_\ell h_{nj} = \sum_k \big(\mathrm{sign}(n - k) h_{nk} h_{kj} + \mathrm{sign}(j - k) h_{kj} h_{nk}\big) \tag{9a}$$

$$\Rightarrow \quad \partial_\ell h_{nn} = 2 \sum_{k \neq n} \mathrm{sign}(n - k) h_{nk} h_{kn}\,. \tag{9b}$$

For finite ($1 \leq n \leq N$) and infinite ($1 \leq n$) matrices we can sum over the first $r$ diagonal elements to show the relation

$$\partial_\ell \left( \sum_{n=1}^{r} h_{nn} \right) = \sum_{n=1}^{r} \sum_{k > r} -2|h_{nk}|^2 \leq 0\,, \tag{10}$$

implying that the sums can only decrease but never increase. Due to the variational principle the quantity $\sum_{n=1}^{r} h_{nn}$ is bounded from below by the sum of the lowest $r$ eigenvalues of $H$

$$\sum_{n=1}^{r} h_{nn} \geq \sum_{n=1}^{r} \lambda_n\,, \tag{11}$$

and thus the flow must converge [32, 50, 65].

## 2.3 Non-Hermitian Flow Equations

The pc-generator can be generalized to non-Hermitian matrices $M$

$$M(\ell) = S(\ell)M(0)S(\ell)^{-1}, \tag{12a}$$

$$S(\ell) := \mathcal{T}_\ell \exp\left(\int_0^\ell \eta(\ell')\mathrm{d}\ell'\right), \tag{12b}$$

with a transformation matrix $S(\ell)$, that needs not necessarily be unitary so that the generator $\eta(\ell)$ needs not necessarily be Antihermitian. For $\ell \to \infty$, we arrive at an effective matrix $M_{\mathrm{eff}}$. The resulting flow equations for $M$ and all relevant observables $O$ read

$$\frac{\mathrm{d}}{\mathrm{d}\ell}M(\ell) = \big[\eta[M(\ell)], M(\ell)\big], \tag{13a}$$

$$\frac{\mathrm{d}}{\mathrm{d}\ell}O(\ell) = \big[\eta[M(\ell)], O(\ell)\big], \tag{13b}$$

in analogy to the Hermitian case. In some cases, these equations can be solved analytically, but in most cases a numerical integration is necessary. Numerical integrations are stopped at a finite $\ell_{\max}$ where the numerical flow is close enough to its converged fixed point $M_{\mathrm{eff}}$. The choice of the generator $\eta$ affects the properties of the flow, in particular the speed with which $M(\ell)$ converges for a given system.

## 2.4 Lindblad Master Equations

As an application of non-Hermitian matrices in physics, we will focus on the Markovian Lindblad master equations

$$\begin{aligned}
i\hbar\frac{\mathrm{d}}{\mathrm{d}t}\rho(t) &= [H, \rho(t)] + i\hbar\sum_\alpha \gamma_\alpha\bigg(L_\alpha\rho(t)L_\alpha^\dagger - \frac{1}{2}(L_\alpha^\dagger L_\alpha\rho(t) + \rho(t)L_\alpha^\dagger L_\alpha)\bigg) \\
&=: \mathcal{L}[\rho(t)],
\end{aligned} \tag{14}$$

which describes the dynamics of a quantum system in contact with an external bath by the time-dependent density matrix $\rho(t)$ and the Hamiltonian $H$. The operators $L_\alpha$ are called Lindblad operators [3], but sometimes also (quantum) jump operators or noise operators [67]. The corresponding rates are denoted by $\gamma_\alpha$. The first summand is the Liouville-von Neumann equation and the other summands describe the coupling (dissipator) to the environment. The Lindbladian $\mathcal{L}$ is a linear superoperator acting on operators in the Fock-Liouville space. For all computational purposes, $\mathcal{L}$ can be represented by a complex matrix, using a basis of $\rho$ in $\mathbb{C}^{(N^2)}$. The Lindbladian $\mathcal{L}$ is not Hermitian and has complex eigenvalues $\lambda$ with $\mathrm{Im}(\lambda) \leq 0$. For all eigenvalues with $\mathrm{Re}(\lambda) \neq 0$, $\mathcal{L}$ also has the eigenvalue $-\lambda^*$. This is due to the symmetry $\mathcal{L} = -\mathcal{L}^\dagger$ of the map. Note that in other references, the Lindbladian is often defined by $\mathcal{L}[\rho(t)] = \frac{\mathrm{d}}{\mathrm{d}t}\rho(t)$, in which case non-real eigenvalues appear in pairs $(\lambda, \lambda^*)$ instead of pairs $(\lambda, -\lambda^*)$. We will consider examples of spectra in $\mathbb{C}$ in Sec. 4.6 as part of the numerical benchmarking.

Since we use the prefactor $i\hbar$ in the definition $\mathcal{L}[\rho(t)] = i\hbar\frac{\mathrm{d}}{\mathrm{d}t}\rho(t)$, the time-evolution of the eigenstates reads $\rho_\lambda(t) = \rho_\lambda(0)\exp(-i\lambda t/\hbar)$. The real parts of the spectrum describe energies, which appear in the temporal evolution as oscillations $\exp(-i\,\mathrm{Re}(\lambda)t/\hbar)$. The imaginary parts describe dissipations. Due to $\mathrm{Im}(\lambda) \leq 0$, all eigenstates decay over time with

$\exp(-|\text{Im}(\lambda)|t/\hbar)$. The eigenvectors to the eigenvalue $\lambda = 0$ corresponds to a steady state $\rho_0(t) = \rho_0$, which can be degenerate. These states do not vanish in the temporal evolution. Quasi-stationary states have an eigenvalue $\lambda$ with $\text{Im}(\lambda) = 0$.

## 2.5 Limitations of the PC-Generator for Non-Hermitian Matrices

For a non-Hermitian matrix $M$ with elements $m_{nj}$, the variational principle does not hold. Furthermore, the flow becomes more intricate. For example, the sum over the first $r$ diagonal flows is

$$\sum_{n=1}^{r} \partial_\ell m_{nn} = \sum_{n=1}^{r} \sum_{k>r} -2m_{nk}m_{kn}, \tag{15}$$

where $m_{nk}m_{kn} \in \mathbb{C}$ needs not be positive or even real, so we cannot derive the inequality (10) that applied for Hermitian matrices $H$.

To understand the flow better, the general matrix $M$ is split into a Hermitian part $H = H^\dagger$ and an Antihermitian part $A = -A^\dagger$

$$M = H + A, \qquad H := \frac{M + M^\dagger}{2}, \quad h_{nj} := (H)_{nj}, \tag{16a}$$

$$A := \frac{M - M^\dagger}{2}, \quad a_{nj} := (A)_{nj}, \tag{16b}$$

where $H$ has a real spectrum and $A$ has an imaginary spectrum. With this, the flow equations read

$$\partial_\ell H = \big[\eta[H], H\big] + \big[\eta[A], A\big], \tag{17a}$$

$$\partial_\ell A = \big[\eta[H], A\big] + \big[\eta[A], H\big]. \tag{17b}$$

The flows of diagonal and non-diagonal components are

$$\partial_\ell h_{nj} = \text{sign}(n - j)\big[h_{nj}(h_{jj} - h_{nn}) + a_{nj}(a_{jj} - a_{nn})\big]$$
$$+ \sum_{k \neq n, j} \big[\text{sign}(n - k)\big(h_{nk}h_{kj} + a_{nk}a_{kj}\big) + \text{sign}(j - k)\big(h_{nk}h_{kj} + a_{nk}a_{kj}\big)\big], \tag{18a}$$

$$\partial_\ell h_{nn} = \sum_{k} 2\text{sign}(n - k)\big(|h_{nk}|^2 - |a_{nk}|^2\big) \in \mathbb{R}, \tag{18b}$$

and

$$\partial_\ell a_{nj} = \text{sign}(n - j)\big[h_{nj}(a_{jj} - a_{nn}) + a_{nj}(h_{jj} - h_{nn})\big]$$
$$+ \sum_{k \neq n, j} \big[\text{sign}(n - k)\big(h_{nk}a_{kj} + a_{nk}h_{kj}\big) + \text{sign}(j - k)\big(a_{nk}h_{kj} + h_{nk}a_{kj}\big)\big], \tag{19a}$$

$$\partial_\ell a_{nn} = \sum_{k} 2\text{sign}(n - k)\big(a_{nk}h_{nk}^* + a_{kn}h_{kn}^*\big) \in i\mathbb{R}. \tag{19b}$$

If the Antihermitian components are significantly smaller than the Hermitian ones, fast convergence can still be observed in our calculations. Many examples of non-Hermitian matrices with real eigenvalues are known for which the pc-generator still convergences. This has been observed, for instance, when using Dyson-Maleev representations of spin observables [47, 48] and for spin lattices subject to a non-Hermitian staggered magnetic field [68].

If the eigenvalues are not real and the Antihermitian components of the matrix are significant, convergence is often not achieved. In App. A we discuss some possible generalizations of the pc-generator to non-Hermitian matrices which prove to be unsuited for real applications. There, we also show the flow and the convergence coefficients of the pc-generator and some generalizations of it for non-Hermitian matrices to illustrate that the pc-generator indeed fails if the Antihermitian part is dominant.

## 2.6 Special Case: Antihermitian Matrix

Before introducing a solution for the general case, let us consider a purely Antihermitian matrix $A$. Such a matrix can easily be expressed by a Hermitian matrix $H$

$$A = -A^\dagger, \tag{20a}$$

$$H := iA = H^\dagger. \tag{20b}$$

For any Hermitian matrix $H$ we have already proven that the pc-generator

$$\eta_{nj}^{\text{pc}}[H] = \text{sign}(n-j)h_{nj}, \tag{21}$$

converges and the diagonal flow $\partial_\ell h_{nn} \in \mathbb{R}$ is real. Evidently, the corresponding generator to use for $A$ is

$$\eta_{nj}^{\text{ipc}}[A] = i\eta_{nj}^{\text{pc}}[H] = \text{sign}(n-j)ia_{nj}, \tag{22}$$

and the corresponding diagonal flow $\partial_\ell a_{nn} \in i\mathbb{R}$ is imaginary. We will call this generator $\eta^{\text{ipc}}$ or *imaginary particle conserving generator (ipc-generator)*. The induced flow of the ipc-generator converges nicely for Antihermitian matrices, but struggles with Hermitian matrices the same way the pc-generator struggles with Antihermitian matrices. Numerical results for the ipc-generator applied to non-Antihermitian matrices can be found in App. A. Note that just like the pc-generator, we can modify this generator slightly, for example to

$$\eta_{nj}^{\text{ipc}}[A] = i\eta_{nj}^{\text{pc}}[H] = \text{sign}(ia_{nn} - ia_{jj})ia_{nj}, \tag{23}$$

if the diagonal elements are not explicitly sorted.

## 2.7 Perturbative Expansion and Truncation Errors

In many cases, the Hilbert space of the physical system is infinite or much too large and the flow equations are not closed or too numerous. In those cases, we introduce a truncation method to reduce the basis to only the most relevant processes. Note that this must not necessarily mean a finite Hilbert space. If we use second quantization, i.e. the basis consists of operator monomials, we only track one coefficients for each operator term and each operator can be represented by an infinite matrix. This way, even a finite amount of operators can fully describe processes on an infinite Hilbert space [49].

A generic and successfull truncation scheme for flow equations relies on a small expansion parameter. One puts the focus on the most significant contributions given by the lowest orders of a perturbative expansion. In order to benchmark such a perturbative truncation we introduce the matrix

$$M = \sum_{n=0}^\infty \lambda_n M^{(n)},$$

with the small perturbation parameter $\lambda < 1$. The Taylor matrices $M^{(n)}$ shall be band-diagonal up to the $n$th minor diagonal, i.e. the off-diagonal elements fulfill $m_{jl}^{(n)} = 0\ \forall |j-l| > n$. For simplification, we truncate high orders $n > n_{\text{max}}$, arriving at

$$M_{\text{trunc}} = \sum_{n=0}^{n_{\text{max}}} \lambda_n M^{(n)}.$$

Note that $M_{\text{trunc}}$ is of band-diagonality $n_{\text{max}}$. We use the notation o$n_{\text{max}}$ for the order, e.g. o2 for a second order expansion including the first two minor diagonals. In the context of flow

equations, we generally choose $n_{\max}$ large enough to not introduce any initial truncation error, i.e. $M(\ell = 0) = M_{\text{trunc}}(\ell = 0)$. During the flow, however, the band-diagonality of the matrix can increase, leading to a truncated matrix

$$M_{\text{trunc}}(\ell) = \sum_{n=0}^{\infty} \lambda_n M^{(n)}(\ell), \qquad \text{with } M^{(n)}(0) = 0 \; \forall n > n_{\max},$$

$$= \sum_{n=0}^{n_{\max}} \lambda_n M^{(n)}(\ell) + \Delta_{\text{M, trunc}}(\ell).$$

We only track terms of maximum order $n_{\max}$ so that the flow equations are still closed in the course of the flow. This leads to a finite truncation error. If we choose a sufficiently large $n_{\max}$ for a given, small perturbation parameter $\lambda$, the truncation introduces a negligibly small truncation error $\Delta_{\text{M, trunc}}$ and the relevant physics is still captured sufficiently well. In this context, it is obvious that the flow should not imply significant contribution on minor diagonals far off. Ideally, it only modifies the tracked diagonal and minor diagonals.

The goal is to choose a generator that minimizes the truncation error, so that the physics is well captured even in low orders. To minimize the error, a renormalizing flow is desirable. Such a renormalizing flow first addresses excitations at large energies before it deals with the ones at low energies. To this end, the speed at which it rotates the matrix elements $m_{nk}$ away scales with $|m_{nn} - m_{kk}|^r$, where the exponent $r$ depends on the chosen generator. Thus, if the diagonal elements are sorted such that $|m_{nn} - m_{kk}| < |m_{nn} - m_{ll}| \; \forall |n - k| < |n - l|$, the renormalizing flow eliminates elements on the far off-diagonals very quickly and slowly moves towards eliminating the elements on the closer off-diagonals. Note that rotating one matrix element renormalizes the other matrix elements. This process leads eventually to the effective model $M_{\text{eff}}$.

If the flow equations do not conserve the band-diagonality of $M$, then rotating away elements close to the diagonal also renormalizes elements far away from the diagonal. In this way, far off-diagonal elements which were initially zero can take a finite value during the flow. If those terms are truncated, however, all renormalizations of them are lost in the calculation. This lost information leads to a significant truncation error $\Delta_{\text{M, trunc}}(\ell)$. This error can still be kept small if the flow is renormalizing, since a renormalizing flow rotates far off-diagonal elements away rapidly, including all renormalizations caused by the slower rotations of close off-diagonal elements. Therefore, no significant renormalization of the far off-diagonals can accumulate and the truncation error $\Delta_{\text{M, trunc}}$ stays small. For this reason, renormalizing generators are desirable for an accurate results in spite of truncation.

For Hermitian matrices the diagonal elements are real and can be ordered unambiguously in absence of degeneracy. For non-Hermitian matrices, however, the diagonal elements are complex and an ordering which fulfills $|m_{nn} - m_{kk}| < |m_{nn} - m_{ll}| \; \forall |n - k| < |n - l|$ does not always exist. Furthermore, this ordering can change the band-diagonality of the matrix, requiring a higher truncation order $n_{\max}$. In the benchmarking in Sec. 4 we consider physical models with and without ordering. We see that reordering is not necessary for every system. But in real applications the challenge of ordering the diagonal elements in a meaningful way must be kept in mind.

# 3 GPC-Generator

## 3.1 Definition

The pc-generator defined in Sec. 2.2 provides a powerful tool for closed quantum system with truncations, since it conserves the band-diagonality and is also renormalizing. However, we

showed in Sec. 2.5 that it fails for non-Hermitian matrices with complex eigenvalues, which appear in open quantum systems. For this reason, we want to generalize this generator scheme to the non-Hermitian case. We saw in Sec. 2.6 that Antihermitian matrices can be treated by the ipc-generator (22), which is the pc-generator with an additional phase factor of $\exp(i\pi/2)$. In a sense, the pc-generator itself already contains a phase factor of $\exp(i0) = 1$. Since the diagonal elements of a Hermitian matrix are real and those of an Antihermitian matrix are imaginary, it is reasonable to assume that for a general matrix $M$ the correct phase factor is directly connected to the complex phase of the diagonal elements $m_{nn} \in \mathbb{C}$. With this idea in mind, we can define the *generalized particle conserving generator (gpc-generator)*

$$\eta_{nj}^{\text{gpc}}[M] = \begin{cases} \frac{m_{nn}^* - m_{jj}^*}{|m_{nn}^* - m_{jj}^*|} m_{nj} & \forall m_{nn} \neq m_{jj}, \\ 0 & \forall m_{nn} = m_{jj}. \end{cases} \tag{24}$$

Note that the prefactor always has an absolute value of 1 and takes the value $\exp(i0) = 1$ for Hermitian matrices $H$ and $\exp(i\pi/2)$ for Antihermitian matrices $A$. These special cases

$$\eta_{nj}^{\text{gpc}}[H] = \text{sign}(h_{nn} - h_{jj})h_{nj} = \eta_{nj}^{\text{pc}}[H] \qquad \forall H = H^\dagger, \tag{25a}$$

$$\eta_{nj}^{\text{gpc}}[A] = i\,\text{sign}(ia_{nn} - ia_{jj})a_{nj} = \eta_{nj}^{\text{ipc}}[A] \qquad \forall A = -A^\dagger, \tag{25b}$$

match the generators (7) and (23) considered before. Note that for non-Hermitian matrices with real eigenvalues, the gpc-generator is also equal to the pc-generator (7), explaining why applying the pc-generator to such matrices leads to convergent flows.

We present other approaches to generalize the pc-generator in App. A and show that they perform unsatisfactorily for non-Hermitian matrices. For this reason, only the gpc-generator will be considered from here on.

## 3.2 Comparison to Generators Suggested Previously

Rosso et al. introduced three generators for dissipative systems [1]. Two of them are motivated by the Wegner generator (8) and generalize it to non-Hermitian matrices. The third generator is inspired by a prior suggestion in the context of quantum chemistry [63]. For clarity, we label the three generators by the superscripts R1, R2 and R3

$$\eta^{\text{R1}} = \left[M^\dagger, M_{\text{nondiag}}\right], \tag{26a}$$

$$\eta^{\text{R2}} = \left[M_{\text{diag}}^\dagger, M_{\text{nondiag}}\right], \quad \eta_{nj}^{\text{R2}} = (m_{nn}^* - m_{jj}^*)m_{nj}, \tag{26b}$$

$$\eta_{nj}^{\text{R3}} = \begin{cases} \frac{m_{nj}}{m_{nn} - m_{jj}}, & \text{if } m_{nn} \neq m_{jj}, \\ 0, & \text{if } m_{nn} = m_{jj}. \end{cases} \tag{26c}$$

By defining $\varphi_{nj}$ as the phase of $m_{nn} - m_{jj}$, a striking relation between three of the four presented generators becomes apparent

$$\eta_{nj}^{\text{R2}}[M] = e^{-i\varphi_{nj}}|m_{nn} - m_{jj}|m_{nj}, \tag{27a}$$

$$\eta_{nj}^{\text{gpc}}[M] = \begin{cases} e^{-i\varphi_{nj}} m_{nj} & \forall m_{nn} \neq m_{jj}, \\ 0 & \forall m_{nn} = m_{jj}, \end{cases} \tag{27b}$$

$$\eta_{nj}^{\text{R3}}[M] = \begin{cases} e^{-i\varphi_{nj}} \frac{1}{|m_{nn} - m_{jj}|} m_{nj} & \forall m_{nn} \neq m_{jj}, \\ 0 & \forall m_{nn} = m_{jj}. \end{cases} \tag{27c}$$

Two of the R-generators use the same phase factor $e^{-i\varphi_{nj}}$ as the gpc-generator, but differ by a factor $|m_{nn} - m_{jj}|$. We can consider all three generators as special cases of a more general definition

$$\eta_{nj}^{(r)}[M] := \begin{cases} e^{-i\varphi_{nj}}|m_{nn} - m_{jj}|^r m_{nj} & \forall m_{nn} \neq m_{jj}, \\ 0 & \forall m_{nn} = m_{jj}, \end{cases} \tag{28a}$$

$$\eta^{(r)}[M] = \begin{cases} \eta^{R2}[M] & r = 1, \\ \eta^{gpc}[M] & r = 0, \\ \eta^{R3}[M] & r = -1. \end{cases} \tag{28b}$$

In this sense, $\eta^{gpc}$ is placed between R2 and R3. Note that we can also define generators with other values of the exponent $r$, including non-integer cases. In this paper we will focus on the three cases $r \in \{-1, 0, 1\}$.

The choice of the generator $\eta$ also affects its physical dimension and hence the dimension of the flow parameter $\ell$. For R1 and R2 the flow parameter has the dimension of an inverse square energy $1/E^2$, for gpc the dimension $1/E$ and for R3 it is without dimension. More generally, $\eta$ has the dimension $E^{r+1}$ and $\ell$ the dimension $1/E^{r+1}$. We also find the exponent $r + 1$ in the asymptotic scaling behavior. We can show this with the flow equations

$$\partial_\ell m_{nj} = \sum_k (\eta_{nk}^{(r)} m_{kj} - m_{nk} \eta_{kj}^{(r)}) \tag{29a}$$

$$= e^{-i\varphi_{nj}}|m_{nn} - m_{jj}|^r m_{nj} m_{jj} - e^{-i\varphi_{nj}}|m_{nn} - m_{jj}|^r m_{nj} m_{nn}$$
$$+ \sum_{k \neq n,j} \left( e^{i\varphi_{kn}}|m_{nn} - m_{kk}|^r + e^{i\varphi_{kj}}|m_{jj} - m_{kk}|^r \right) m_{nk} m_{kj} \tag{29b}$$

$$= -\underbrace{|m_{nn} - m_{jj}|^{r+1}}_{=:|\Delta E_{nj}|^{r+1}} \underbrace{m_{nj}}_{\mathcal{O}(M_{\text{nondiag}})}$$
$$+ \sum_{k \neq n,j} \left( e^{i\varphi_{kn}}|m_{nn} - m_{kk}|^r + e^{i\varphi_{kj}}|m_{jj} - m_{kk}|^r \right) \underbrace{m_{nk} m_{kj}}_{\mathcal{O}(M_{\text{nondiag}}^2)} . \tag{29c}$$

We call the differences of the diagonal elements $\Delta E$ in analogy to flow equations for Hamiltonians, where the eigenvalues are energies. We see that for large $\ell$, assuming that the matrix is already close to its diagonal form, the first term dominates. Since, in this case, the diagonals have already converged up to first order in $M_{\text{nondiag}}$ one has $m_{nn}(\ell) = m_{nn}(\infty) + \mathcal{O}(M_{\text{nondiag}}^2)$. Hence, the off-diagonals show an asymptotic convergence $m_{nj} \propto \exp(-|\Delta E_{nj}|^{r+1} \ell)$ with $\Delta E_{nj} = m_{nn}(\infty) - m_{jj}(\infty)$.

Explicitly, the flow equations for R2 ($r = 1$), gpc ($r = 0$) and R3 ($r = -1$) read

$$\text{R2:} \ \partial_\ell m_{nj} = -\underbrace{|m_{nn} - m_{jj}|^2}_{|\Delta E_{nj}|^2} \underbrace{m_{nj}}_{\mathcal{O}(M_{\text{nondiag}})} + \sum_{k \neq n,j} \left( m_{nn}^* + m_{jj}^* - 2m_{kk}^* \right) \underbrace{m_{nk} m_{kj}}_{\mathcal{O}(M_{\text{nondiag}}^2)} , \tag{30a}$$

$$\text{gpc:} \ \partial_\ell m_{nj} = -\underbrace{|m_{nn} - m_{jj}|}_{|\Delta E_{nj}|} \underbrace{m_{nj}}_{\mathcal{O}(M_{\text{nondiag}})} + \sum_{k \neq n,j} \left( \frac{m_{nn}^* - m_{kk}^*}{|m_{nn}^* - m_{kk}^*|} + \frac{m_{jj}^* - m_{kk}^*}{|m_{jj}^* - m_{kk}^*|} \right) \underbrace{m_{nk} m_{kj}}_{\mathcal{O}(M_{\text{nondiag}}^2)} , \tag{30b}$$

$$\text{R3:} \ \partial_\ell m_{nj} = -\underbrace{m_{nj}}_{\mathcal{O}(M_{\text{nondiag}})} + \sum_{k \neq n,j} \left( \frac{1}{m_{nn} - m_{kk}} + \frac{1}{m_{jj} - m_{kk}} \right) \underbrace{m_{nk} m_{kj}}_{\mathcal{O}(M_{\text{nondiag}}^2)} . \tag{30c}$$

Note that the asymptotic convergence and the dimensions of the generators are the same as those of the original generator they are based on, i.e. the Wegner generator for R2, the pc-generator for gpc and the White generator for R3. Furthermore, the asymptotic convergence

of gpc and R2 scales with the energy differences, which makes their flow renormalizing. We emphasize that this is a desirable property for minimizing the truncation error as discussed in Sec. 2.7. The gpc-generator does not, however, preserve the band-diagonality of $M$ like the pc-generator does for Hermitian matrices, see App. B. While R1 does not directly fit into the more general definition (28), it shows a linear energy dependence, just like R2, and we will see in our results that both generators perform similarly. Only R3 is not renormalizing, because it deals with all energy differences simultaneously.

The units and convergence behavior of the generators are summarized in Tab. 1.

Table 1: Comparison of the dimensions and convergence behaviours of the generators considered in this work.

| Generator | $[\eta]$ | $[\ell]$ | Convergence behavior |
|---|---|---|---|
| R1, R2 | $E^2$ | $1/E^2$ | $\exp(-\ell|\Delta E|^2)$ |
| gpc | $E$ | $1/E$ | $\exp(-\ell|\Delta E|)$ |
| R3 | $1$ | $1$ | $\exp(-\ell)$ |
| $\eta^{(r)}$ | $E^{1+r}$ | $1/E^{1+r}$ | $\exp(-\ell|\Delta E|^{1+r})$ |

## 3.3 Proof of Convergence

We show convergence of the flow induced by the gpc-generator using perturbative arguments in a general expansion parameter $x$. We assume a matrix without degeneracy (for simplicity) with non-diagonal elements that can be expanded in a Taylor series

$$m_{nn}(\ell) = m_n(\ell), \qquad m_a(\ell) \neq m_b(\ell) \ \forall a \neq b, \tag{31a}$$

$$m_{nl}(\ell) = \sum_{i>0} a_{nl}^{(i)}(\ell) x^i, \quad n \neq l, \tag{31b}$$

where $|x| < 1$ is the expansion parameter. For brevity of notation, we define

$$c_{nkl} := \left( \frac{m_n^* - m_k^*}{|m_n^* - m_k^*|} + \frac{m_l^* - m_k^*}{|m_l^* - m_k^*|} \right) \Rightarrow |c_{nkl}| \leq 2 \tag{32}$$

and calculate the flow of the non-diagonal elements

$$\partial_\ell \left( \sum_{i>0} a_{nl}^{(i)}(\ell) x^i \right) = -|m_n - m_l| \left( \sum_{i>0} a_{nl}^{(i)} x^i \right) \tag{33a}$$

$$+ \sum_{k \neq n,l} c_{nkl} \left( \sum_{i>0} a_{nk}^{(i)} x^i \right) \left( \sum_{j>0} a_{kl}^{(j)} x^j \right), \tag{33b}$$

$$\Rightarrow \partial_\ell a_{nl}^{(i)}(\ell) = -|m_n - m_l| a_{nl}^{(i)} + \sum_{k \neq n,l} c_{nkl} \left( \sum_{0 < \delta < i} a_{nk}^{(\delta)} a_{kl}^{(i-\delta)} \right). \tag{33c}$$

The $\ell$-dependence is not denoted explicitly for brevity of notation, but all $a_{nk}^{(i)}$ still depend on $\ell$. We show convergence using induction in $i$ by proving that all orders $j \leq i$ of the non-diagonal elements $m_{nl}(\ell)$ become exponentially small beyond a value $\ell_i$, i.e.

$$|a_{nl}^{(j)}(\ell)| \ll 1, \qquad \forall i \geq j \geq 0; \ell > \ell_i. \tag{34}$$

The induction basis is

$$a_{nl}^{(0)}(\ell) = 0, \qquad \forall \ell > \ell_0, \tag{35}$$

which is fulfilled with $\ell_0 = 0$ since the constant order $a_{nl}^{(0)}(\ell)$ vanishes for all non-diagonal elements by definition of the initial conditions. For the induction step, we assume that all orders $j < i$ have become exponentially small

$$|a_{nl}^{(j)}(\ell)| \ll 1, \qquad \forall j < i; \ \ell > \ell_{\max,\,i} := \max(\ell_0, ... \ell_{i-1}). \qquad (36)$$

Then, obviously $a_{nk}^{(\delta)} a_{kl}^{(i-\delta)} \lll 1$ for all $\delta \in \mathbb{N}$ that fulfill $0 < \delta < i$. With $|c_{nkl}| \le 2$ the second summand in (33c) becomes negligible compared to the first summand. Therefore, we obtain

$$\partial_\ell a_{nl}^{(i)}(\ell) \approx -|m_n - m_l| a_{nl}^{(i)}(\ell), \qquad \forall \ell > \ell_{\max,\,i}, \qquad (37a)$$

$$\Rightarrow \quad a_{nl}^{(i)}(\ell) \approx a_{nl}^{(i)}(\ell_{\max,\,i}) e^{-|m_n(\infty) - m_l(\infty)|\ell}, \qquad \forall \ell > \ell_{\max,\,i}, \qquad (37b)$$

$$\Rightarrow \quad a_{nl}^{(i)}(\ell) \ll 1, \qquad \forall \ell > \ell_i \text{ with } \ell_i \gg \ell_{\max,\,i}. \qquad (37c)$$

If $\ell_i$ is chosen large enough, $a_{nl}^{(i)}(\ell) \ll 1 \ \forall \ell > \ell_i$ is fulfilled. This concludes the induction step. Note that no specific value for $\ell_i$ is given in the derivation. The convergence speed depends on the matrix elements $m_n(\ell)$ and $m_l(\ell)$, but for an arbitrarily large $\ell_i$, convergence will eventually be achieved. $\square$

    We point out that the series in $x$ can be ill-behaved if $\ell_i$ grows quickly with $i$ so that the series $m_{nl}(\ell)$ does not converge in spite of the convergence of all $a_{nl}^{(i)}$. In all our numerical calculations, however, we observed that the gpc-generator induces a convergent flow.

## 4 Benchmark Results

We want to compare the flow induced by the generators analytically in Sec. 4.2 and numerically in Secs. 4.3-4.6. For the numerical benchmarking, two criteria are of interest: The convergence speed in real time and the truncation error, which are measured separately. Note that often rapid convergence and minimal truncation errors are two orthogonal requirements and that it is important to choose a generator scheme which represents a good compromise.

### 4.1 Benchmark Parameters

**Convergence Speed**

Rapid convergence is advantageous because it allows for fast numerical calculations for complicated systems. We need a quantity to measure the convergence of the flow. For this, we use the *residual-off-diagonality (ROD)* [46, 66]

$$\boxed{\mathrm{ROD}[M] = \frac{1}{D} \sqrt{\sum_i \sum_{j \neq i} |m_{ij}|^2},} \qquad (38)$$

where $D$ is the dimension of the matrix $M$. The ROD measures the size of all off-diagonal matrix elements and converges to 0 while the matrix converges to a fixed point of the flow. In our benchmark tests, we always diagonalize the matrix which implies that the ROD indeed vanishes. It should be noted, however, that in most physical applications a full diagonalization is not desired and the ROD can be adapted to sum only over the matrix elements that should vanish in $M_{\mathrm{eff}} = M(\ell \to \infty)$ [66] in order to simplify the problem at hand. We stop the integration when $\mathrm{ROD} < 10^{-8} J$ is fulfilled, where $J = [m_{00}]$ is the energy dimension of the matrix. For numerical calculations with non-physical matrices, we simply set $J$ to 1. The ROD

is often used as a convergence measure in applications of flow equations where the spectrum is usually not known beforehand.

To display the convergence behavior of the function $\text{ROD}(\ell)$ concisely we introduce the *convergence coefficient*

$$C_{\text{Conv}}^{(\ell)} = -\frac{\ln[\text{ROD}(\ell_{\max})/\text{ROD}(\ell_{\min})]}{\ell_{\max} - \ell_{\min}} . \tag{39}$$

It measures the speed of the exponential convergence of the ROD according to the asymptotic behavior $\text{ROD} \propto \exp(-C_{\text{Conv}}^{(\ell)}\ell)$. The larger $C_{\text{Conv}}^{(\ell)}$, the faster the flow converges. The upper limit $\ell_{\max}$ is chosen such that $\text{ROD}(\ell_{\max}) \approx 1\text{e-}6\,J$ and $\ell_{\min}$ such that $\text{ROD}(\ell_{\min}) \approx 2\,\text{ROD}(\ell_{\max})$ to ensure that initial transient behavior at $\ell \approx 0$ as well as irrelevant fluctuations are ignored. If no convergence is achieved, $C_{\text{Conv}}^{(\ell)}$ is set to 0.

While $C_{\text{Conv}}^{(\ell)}$ is a natural first choice, the generators induce different energy scales and thus different dimensions for $\ell$, which makes a direct comparison difficult. For this reason we focus on the variant

$$\boxed{C_{\text{Conv}}^{(t)} = -\frac{\ln[\text{ROD}(t_{\max})/\text{ROD}(t_{\min})]}{t_{\max} - t_{\min}},} \tag{40}$$

using the actual computation time $t$. Measuring the performance in real time is prone to other artifacts, e.g. increased computation times due to throttling of the CPU. While this can introduce random fluctuations in the measured times, we observe that the temporal coefficient $C_{\text{Conv}}^{(t)}$ still gives a more meaningful benchmark for the real-life performance of the various generators than $C_{\text{Conv}}^{(\ell)}$.

In some cases we see that $C_{\text{Conv}}^{(t)}$ fails to capture the flow since the flow $\text{ROD}(t)$ deviates too much from an exponential decay. For this reason we also show exemplary plots of the full flow $\text{ROD}(t)$. We also check all our results for correct convergence by comparing the diagonal elements once the flow equations have converged with the eigenvalues obtained by standard diagonalization.

**Truncation Error**

Rapid convergence speed is only desirable if it does not entail large truncation errors. To benchmark the accuracy of the generators in spite of truncation, we initialize a matrix $M_{\text{prep}}$ based on a physical or mathematical model $M$. For the preparation, we truncate $M$ in order $on_{\max}$ so that $M_{\text{prep}}$ is band-diagonal with diagonal width $n_{\max}$, see Sec. 2.7. This truncation scheme is reasonable if the truncated off-diagonal elements are small. If this is not true for a given model $M$ in our benchmark, we introduce the expansion parameter $\lambda$ as before according to $m_{nj,\,\text{prep}} = \lambda^{|n-j|} m_{nj}$. Another useful step in the initialization of $M_{\text{prep}}$ is the reordering of the diagonals, so that the condition $|m_{nn} - m_{jj}| > |m_{nn} - m_{kk}|$ for $|n-j| > |n-k|$ is fulfilled. Note that we always perform the truncation to a band-diagonal matrix, but we only introduce the expansion parameter or reordering if we mention this in the results for the specific model. As an instructive example of perturbative truncation based on $\lambda$ consider a 4x4 matrix in truncation order o2

$$M = \begin{pmatrix} m_{11} & m_{12} & m_{13} & m_{14} \\ m_{21} & m_{22} & m_{23} & m_{24} \\ m_{31} & m_{32} & m_{33} & m_{34} \\ m_{41} & m_{42} & m_{43} & m_{44} \end{pmatrix} \Rightarrow M_{\text{prep}} = \begin{pmatrix} m_{11} & \lambda m_{12} & \lambda^2 m_{13} & 0 \\ \lambda m_{21} & m_{22} & \lambda m_{23} & \lambda^2 m_{24} \\ \lambda^2 m_{31} & \lambda m_{32} & m_{33} & \lambda m_{34} \\ 0 & \lambda^2 m_{42} & \lambda m_{43} & m_{44} \end{pmatrix}. \tag{41}$$

After initializing $M_{\text{prep}}$ we solve the flow equations in order $on_{\max}$ to calculate the spectrum $\Lambda_{\text{trunc}}$. Any element $m_{nj}(\ell)$ with $|n-j| > n_{\max}$ is truncated during the whole calculation,

so truncation errors arise and the size of these errors depends on the model and the used generator. We compare $\Lambda_{\text{trunc}}$ with the spectrum $\Lambda_{\text{exact}}$ of $M_{\text{prep}}$ obtained with standard diagonalization

$$M_{\text{prep}}, \quad \text{with spectrum } \Lambda_{\text{exact}} = (\lambda_{1,\,\text{exact}}, \lambda_{2,\,\text{exact}}, ..., \lambda_{D,\,\text{exact}}), \tag{42a}$$

$$\xrightarrow{\text{flow equations}} M_{\text{eff}}, \quad \text{with spectrum } \Lambda_{\text{trunc}} = (\lambda_{1,\,\text{trunc}}, \lambda_{2,\,\text{trunc}}, ..., \lambda_{D,\,\text{trunc}}), \tag{42b}$$

$$\Delta_{\text{trunc}} = \left| \Lambda_{\text{trunc}} - \Lambda_{\text{exact}} \right| = \sqrt{\sum_n \left| \lambda_{i,\,\text{trunc}} - \lambda_{i,\,\text{exact}} \right|^2}. \tag{42c}$$

Note that $M_{\text{prep}}$ is prepared specifically to compare the truncation errors of different generators with one another, not to compare the error of different truncation orders. Since the initial matrix $M_{\text{eff}}(0) = M_{\text{prep}}$ is already truncated in order on$_{\text{max}}$, it is less scarce in higher orders, so the error $\Delta_{\text{trunc}}$ does not always decrease upon increasing the order. In real applications, one considers a fixed initial matrix and can decrease the truncation error by increasing $n_{\text{max}}$.

We implement the calculations in C++ and use the Eigen library [69] for matrix arithmetics and the Runge-Kutta-Dopri5 algorithm from the Boost library [70] for integrating the flow equations. This integration algorithm uses a controlled stepper that adjusts the stepsize $\Delta \ell$ to ensure a maximum absolute and relative error of $10^{-8}$ of each step. All computations are performed on the same machine and the computational runtimes are measured in seconds. Since the time depends strongly on the used hardware and software optimizations, absolute values are not representative, but the relative values provide information on the performance of the generators.

## 4.2 Analytical Example: Fermionic Mode with Losses and Gains

### 4.2.1 Physical Model

We first discuss a purely analytical example [1] to acquire a basic understanding of the flow equations. Consider a system with a single fermionic mode of energy $\epsilon$ coupled to a bath with loss rate of $\Gamma_1$ and gain rate of $\Gamma_2$. The Lindblad operators are the canonical fermionic creation and annihilation operators $\hat{c}$ and $\hat{c}^\dagger$. The Lindblad master equations read

$$i\hbar \frac{\mathrm{d}}{\mathrm{d}t} \rho(t) = [H, \rho(t)] + i\hbar \sum_j \left( \Gamma_j L_j \rho(t) L_j^\dagger - \frac{1}{2} \Gamma_j \left\{ L_j^\dagger L_j, \rho(t) \right\} \right), \tag{43a}$$

$$\hat{H} = \epsilon \hat{c}^\dagger \hat{c}, \quad L_1 = \hat{c}, \quad L_2 = \hat{c}^\dagger. \tag{43b}$$

We want to treat the fermionic problem including the bath as a fermionic two-level problem. To do so, we study this model using the *superfermion representation* (presented in more detail in Refs. [1,71,72]), where we express the density matrix $\rho = \sum_{nm} \rho_{nm} |n\rangle \langle m|$ by a vector in Fock-Liouville space

$$|\rho\rangle = \sum_{nm} \rho_{nm} |n\rangle \otimes |m\rangle = \rho \otimes \mathbb{1} |I\rangle, \tag{44a}$$

$$|I\rangle := \sum_n |n\rangle \otimes |n\rangle. \tag{44b}$$

Note that $|I\rangle$ is the expression of the identity matrix in Fock-Liouville space. Now, we can define fermionic superoperators

$$c = \hat{c} \otimes \mathbb{1}, \tag{45a}$$

$$\tilde{c} = (-1)^{c^\dagger c} \otimes \hat{c}, \tag{45b}$$

which describe the action of the fermionic operators $\hat{c}$ and $\hat{c}^\dagger$ on the left side or on the right side of the density matrix. These operators fulfill the fermionic anticommutation relations

$$\{c,c\} = 0, \qquad\qquad \{c,c^\dagger\} = 1, \tag{46a}$$

$$\{\tilde{c},\tilde{c}\} = 0, \qquad\qquad \{\tilde{c},\tilde{c}^\dagger\} = 1, \tag{46b}$$

$$\{c,\tilde{c}\} = 0, \qquad\qquad \{c,\tilde{c}^\dagger\} = 0. \tag{46c}$$

Applying the formalism yields the master equations in the form

$$i\hbar \frac{d}{dt}|\rho(t)\rangle = M|\rho(t)\rangle, \tag{47}$$

where $M$ is a bilinear expression

$$M = \begin{pmatrix} c^\dagger & \tilde{c} \end{pmatrix} \begin{pmatrix} \epsilon - \frac{i\hbar}{2}\Delta\Gamma_{12} & \hbar\Gamma_2 \\ -\hbar\Gamma_1 & \epsilon + \frac{i\hbar}{2}\Delta\Gamma_{12} \end{pmatrix} \begin{pmatrix} c \\ \tilde{c}^\dagger \end{pmatrix} - \epsilon - \frac{i\hbar}{2}(\Gamma_1 + \Gamma_2), \tag{48}$$

that can be diagonalized by a non-unitary, but invertible transformation $M_{\text{eff}} = SMS^{-1}$ with the diagonal elements $\lambda_1$ and $\lambda_2$. We define new operators

$$\begin{pmatrix} d \\ \tilde{d}^\dagger \end{pmatrix} = S \begin{pmatrix} c \\ \tilde{c}^\dagger \end{pmatrix}, \qquad (D^\dagger, \tilde{D}) = (c^\dagger, \tilde{c})S^{-1}, \tag{49}$$

which also satisfy canonical anticommutation relations. Note that $(d)^\dagger \neq D^\dagger$ and $(\tilde{D})^\dagger \neq \tilde{d}^\dagger$, but the operators fulfill the relations $\{d, D^\dagger\} = 1$ and $\{\tilde{d}^\dagger, \tilde{D}\} = 1$. We obtain the diagonal matrix

$$M_{\text{eff}} = \begin{pmatrix} D^\dagger & \tilde{D} \end{pmatrix} \begin{pmatrix} \epsilon - \frac{i\hbar}{2}(\Gamma_1 + \Gamma_2) & 0 \\ 0 & \epsilon + \frac{i\hbar}{2}(\Gamma_1 + \Gamma_2) \end{pmatrix} \begin{pmatrix} d \\ \tilde{d}^\dagger \end{pmatrix} - \epsilon - \frac{i\hbar}{2}(\Gamma_1 + \Gamma_2), \tag{50}$$

with $\Delta\Gamma_{12} := \Gamma_1 - \Gamma_2$. With this, we obtain the analytical result

$$\lambda_\pm = \epsilon \pm \frac{i\hbar}{2}(\Gamma_1 + \Gamma_2). \tag{51}$$

We gauge the flow equation results with this analytical expression. We conduct the diagonalization of the $(2 \times 2)$-matrix with the presented generators of the flow equation parametrizing the Lindbladian matrix as

$$M(\ell) = \begin{pmatrix} \epsilon(\ell) + i\alpha(\ell) & \mu_2(\ell) \\ -\mu_1(\ell) & \epsilon(\ell) - i\alpha(\ell) \end{pmatrix}, \quad \alpha(\ell), \epsilon(\ell), \mu_{1,2}(\ell) \in \mathbb{R}, \tag{52a}$$

$$\alpha(0) = -\frac{\hbar}{2}\Delta\Gamma_{12}, \qquad \mu_{1,2}(0) = \hbar\Gamma_{1,2}, \qquad \epsilon(0) = \epsilon. \tag{52b}$$

We stress that the matrix representation $M(\ell)$ fully determines the bilinear operator terms of (48) in second quantization.

### 4.2.2  Flow Equations and Stationary State

The gpc-generator for this problem reads

$$\eta^{\text{gpc}} = \begin{pmatrix} 0 & \frac{(\epsilon - i\alpha) - (\epsilon + i\alpha)}{|(\epsilon - i\alpha) - (\epsilon + i\alpha)|}\mu_2 \\ \frac{(\epsilon + i\alpha) - (\epsilon - i\alpha)}{|(\epsilon + i\alpha) - (\epsilon - i\alpha)|}(-\mu_1) & 0 \end{pmatrix} = \begin{pmatrix} 0 & -i\mu_2\text{sign}(\alpha) \\ -i\mu_1\text{sign}(\alpha) & 0 \end{pmatrix}, \tag{53}$$

and the flow equations read

$$\partial_\ell M = [\eta^{\text{gpc}}, M]\tag{54a}$$

$$= \begin{pmatrix} \text{i}2\mu_1\mu_2\text{sign}(\alpha) & -2\mu_2|\alpha| \\ 2\mu_1|\alpha| & -\text{i}2\mu_1\mu_2\text{sign}(\alpha) \end{pmatrix},\tag{54b}$$

yielding the following differential equations for the parameters

$$\text{gpc: } \partial_\ell\alpha = 2\mu_1\mu_2\text{sign}(\alpha), \qquad \partial_\ell\mu_1 = -2\mu_1|\alpha|, \qquad \partial_\ell\mu_2 = -2\mu_2|\alpha|.\tag{55a}$$

For the other generators the following differential equations result

$$\text{R1: } \partial_\ell\alpha = 4\mu_1\mu_2\alpha, \qquad \partial_\ell\mu_1 = -2\mu_1(2\alpha^2 + \mu_1^2 - \mu_2^2), \qquad \partial_\ell\mu_2 = -2\mu_2(2\alpha^2 - \mu_1^2 + \mu_2^2),\tag{56a}$$

$$\text{R2: } \partial_\ell\alpha = 4\mu_1\mu_2\alpha, \qquad \partial_\ell\mu_1 = -4\mu_1\alpha^2, \qquad \partial_\ell\mu_2 = -4\mu_2\alpha^2,\tag{56b}$$

$$\text{R3: } \partial_\ell\alpha = \mu_1\mu_2/\alpha, \qquad \partial_\ell\mu_1 = -\mu_1, \qquad \partial_\ell\mu_2 = -\mu_2.\tag{56c}$$

While all of these differential equations are quite simple, the system of differential equations for R1 is slightly lengthier and the equations for the gpc-generator seem slightly more complicated than the ones for R2 and R3 due to the sign of $\alpha$. By using $|\alpha|$ instead of $\alpha$ we can reduce the complexity of the equations for the gpc-generator.

By using the two invariants of motion

$$\text{Tr}[M] = 2\epsilon = \text{const.} \qquad \Rightarrow \qquad \epsilon = \text{const},\tag{57a}$$

$$\text{Tr}[M^2] = 2(\epsilon^2 - \alpha^2) - 2\mu_1\mu_2 = \text{const.} \qquad \Rightarrow \qquad \alpha^2 + \mu_1\mu_2 = \hbar^2\frac{(\Gamma_1 + \Gamma_2)^2}{4},\tag{57b}$$

we obtain a single reduced equation of motion for $\alpha$ for each generator

$$\text{gpc: } \partial_\ell\alpha(\ell) = \frac{1}{2}\left(\hbar^2(\Gamma_1 + \Gamma_2)^2 - 4\alpha^2(\ell)\right)\text{sign}(\alpha),\tag{58a}$$

$$\text{R1,R2: } \partial_\ell\alpha(\ell) = \left(\hbar^2(\Gamma_1 + \Gamma_2)^2 - 4\alpha^2(\ell)\right)\alpha(\ell),\tag{58b}$$

$$\text{R3: } \partial_\ell\alpha(\ell) = \frac{\hbar^2(\Gamma_1 + \Gamma_2)^2}{4\alpha(\ell)} - \alpha(\ell),\tag{58c}$$

where the different units of the flow parameter $\ell$ can be seen clearly because $\alpha$, $\mu_i$ and $\hbar\Gamma_i$ are energies. The first three generators exhibit three fixed points at $\tilde{\alpha}_{1,2} = \pm\hbar(\Gamma_1 + \Gamma_2)/2$ and $\tilde{\alpha}_3 = 0$. We check the stability of the fixed point by differentiating the flow equation $\partial_\ell\alpha =: f(\alpha)$ respectively, receiving $f'(\alpha)$ and evaluating it at the fixed points. Since $f'_{\text{R3}}(\alpha)$ diverges at $\alpha = 0$, we evaluate the sign of $\partial_\ell$ in the proximity of $\tilde{\alpha}_3$ directly

$$f'_{\text{gpc}}(\tilde{\alpha}_{1,2}) = -2\hbar|\Gamma_1 + \Gamma_2| \leq 0, \qquad f'_{\text{gpc}}(\tilde{\alpha}_3) = \hbar^2(\Gamma_1 + \Gamma_2)^2\delta(0) \geq 0,\tag{59a}$$

$$f'_{\text{R1,R2}}(\tilde{\alpha}_{1,2}) = -2\hbar^2(\Gamma_1 + \Gamma_2)^2 \leq 0, \qquad f'_{\text{R1,R2}}(\tilde{\alpha}_3) = (\Gamma_1 + \Gamma_2)^2 \geq 0,\tag{59b}$$

$$f'_{\text{R3}}(\alpha) < 0 \; \forall\alpha \neq 0, \qquad \text{sign}\left(\partial_\ell\alpha_{\text{R3}}(\ell)\right) = \text{sign}(\alpha_{\text{R3}}) \text{ for } \alpha_{\text{R3}} \approx \tilde{\alpha}_3 = 0.\tag{59c}$$

We see that for all generators the fixed points $\tilde{\alpha}_{1,2}$ are stable and the fixed point $\tilde{\alpha}_3$ is unstable. This example shows that all four generators exhibit the same stationary value behavior, even though the prefactors of the flow equations and therefore the convergence speeds in $\ell$ differ which can be seen in Fig. 1. Recall that $\ell$ has different units for the generators, so that the above observation is rather a statement about energy dependence than about real-time convergence speed. An analytical solution of the flow equations for the gpc-generator and a simple observable are derived in App. C.

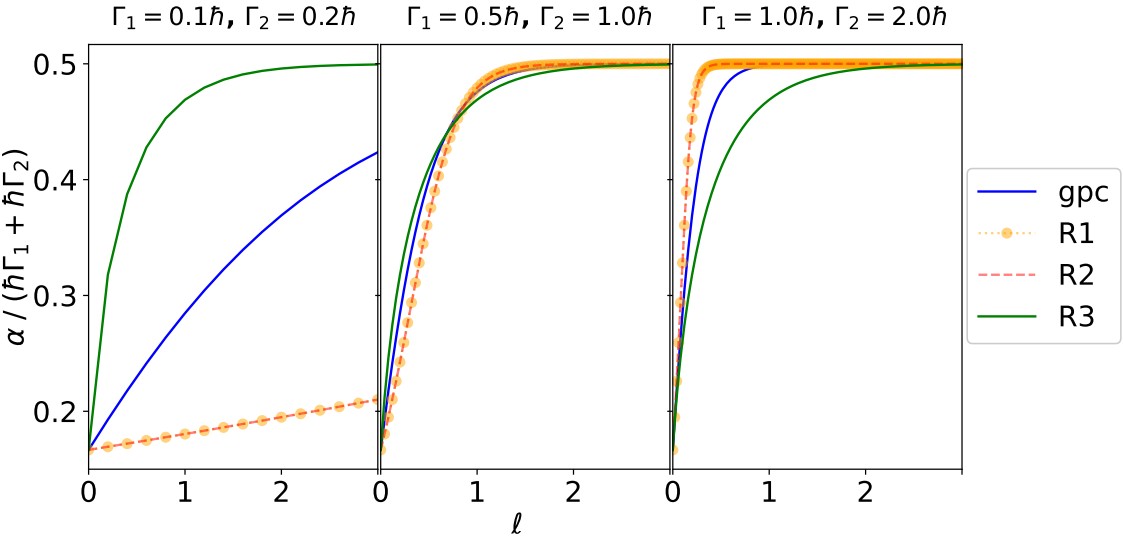

Figure 1: Exemplary flows $\alpha(\ell)$ of the analytical example discussed in Sec. 4.2.

### 4.2.3 Simple Analytical Case $\Gamma_2 = 0$

We consider the special case $\Gamma_2 = 0$ to further analyze the differences between the generators. In this case, $\mu_2(\ell) = 0$ and by using the second invariant we also obtain $\alpha(\ell) = -\Gamma_1/2$. By using the second invariant in the differential equation for the remaining degree of freedom $\mu_1(\ell)$, we obtain

$$\text{gpc: } \partial_\ell \mu_1(\ell) = -\hbar\Gamma_1 \mu_1(\ell), \tag{60a}$$

$$\text{R1: } \partial_\ell \mu_1(\ell) = -2(\hbar^2\Gamma_1^2 + \mu_1^2(\ell))\mu_1(\ell), \tag{60b}$$

$$\text{R2: } \partial_\ell \mu_1(\ell) = -\hbar^2\Gamma_1^2 \mu_1(\ell), \tag{60c}$$

$$\text{R3: } \partial_\ell \mu_1(\ell) = -\mu_1(\ell), \tag{60d}$$

with $\mu_1(0) = \hbar\Gamma_1$. Again, we see the different units of $\ell$ clearly. This example also demonstrates nicely how the R3-generator does not depend on the energy scales of the system, since $\Gamma_1$ does not appear as a prefactor in the corresponding flow equation for $\mu_1$. The gpc-generator depends on the energy linearly while R1 and R2 show a quadratic dependence. The flow equations can be solved analytically

$$\mu_1^{\text{gpc}}(\ell) = \hbar\Gamma_1 \exp(-\hbar\Gamma_1\ell), \tag{61a}$$

$$\mu_1^{\text{R1}}(\ell) = \frac{\hbar\Gamma_1}{\sqrt{2\exp(4\hbar^2\Gamma_1^2\ell) - 1}}, \tag{61b}$$

$$\mu_1^{\text{R2}}(\ell) = \hbar\Gamma_1 \exp(-\hbar^2\Gamma_1^2\ell), \tag{61c}$$

$$\mu_1^{\text{R3}}(\ell) = \hbar\Gamma_1 \exp(-\ell). \tag{61d}$$

From this example we see that all generators succeed in the diagonalization. There are differences in the convergence speed which slightly favor R3. No statement on truncation errors is possible since all flows can be computed without approximations.

## 4.3 Random Matrices

### 4.3.1 Matrix Generation

We now want to benchmark both the convergence speed and truncation errors for various non-Hermitian matrices. To cover a variety of matrices without bias, we generate random

($D$x$D$)-matrices with various ratios of Hermitian and Antihermitian contributions. To check the influence of Hermiticity and Antihermiticity of the matrices on the flow, we introduce a crossover ratio $\alpha$. First, we construct a matrix $R$ by sampling the real part and imaginary part of all elements $m_{nj}$ from a uniform distribution on the interval $[-1,1]$. We use the uniform distribution instead of other, unbounded distributions such as the normal distribution to avoid the occurrence of extremely large matrix elements. Then, we construct

$$M := (1-\alpha)(R+R^\dagger) + \alpha(R-R^\dagger) = R + (1-2\alpha)R^\dagger, \tag{62}$$

such that for $\alpha = 0$ the matrix $M$ is Hermitian and for $\alpha = 1$ it is Antihermitian.

### 4.3.2 Convergence Speed

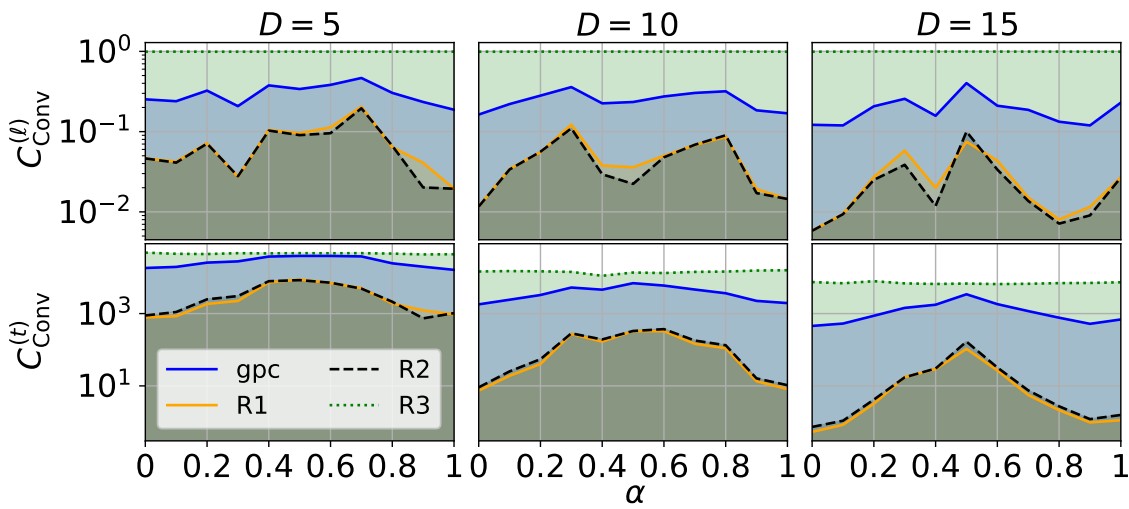

Figure 2: Convergence coefficients for $\ell$-dependent (first row) and time-dependent (second row) flow for $D$-dimensional random matrices, see Sec. 4.3.2, for various crossover ratios $\alpha$ induced by the gpc-generator compared to the flows induced by the R1-, R2- and R3-generator averaged over 100 samples. The dimension $D$ is given at the top of each column.

The bottom row of Fig. 2 shows the convergence coefficients $C_{\text{Conv}}^{(t)}$ for ROD($t$) averaged over 100 samples of random matrices for various matrix dimensions $D$. We use the same 100 random matrices for all four generators in order to compare the performance fairly. Recall that a larger coefficient stands for faster convergence, thus is favorable. Clearly, the flows induced by the gpc-generator and R3-generator converge much more quickly than the ones induced by the R1- and R2-generator with the R3-generator slightly outperforming the gpc-generator.

It also seems that gpc, R1 and R2 perform better for matrices with $\alpha \approx 0.5$ with the worst performance for $\alpha \in \{0,1\}$, when $M$ is completely Hermitian or Antihermitian. One possible explanation for this effect is the distribution of the diagonal elements in the complex plane. For $\alpha = 0$ they are distributed randomly on a line with $\text{Im}(m_{nn}(\ell)) = 0$ and for $\alpha = 1$ they lie on a line with $\text{Im}(m_{nn}(\ell)) = 1$. They stay on these lines during the entire calculation. This makes it more likely for two diagonal elements to be very close to one another, slowing down the renormalizing flows. For $\alpha = 0.5$ the eigenvalues are distributed on a two-dimensional sphere, where diagonal elements are less likely to be very close to each other. Hence the flows converge faster. We created histograms (not shown) of all diagonal differences for various $\alpha$ confirming this hypothesis. We point out that this observation favors the renormalizing generators. Their disadvantage is the fact that they sacrifice convergence speed because they

do not treat all matrix elements at the same speed compared to the R3-generator which treats all elements simultaneously. Our results suggest that this disadvantage is less pronounced for matrices with a strong mixture of Hermitian and Antihermitian parts.

For sake of completeness, we also show $C_{\text{Conv}}^{(\ell)}$ for ROD($\ell$) in the top row of Fig. 2. Here, the R3-generator exhibits a constant $C_{\text{Conv}}^{(\ell)} = 1$ that is larger than the convergence coefficients of the other generators. This is expected, since the scale of $\Delta E$ does not matter for the R3 flow as discussed in Sec. 3.2. We can discern that the fluctuations of $C_{\text{Conv}}^{(\ell)}$ for the generators gpc, R1 and R2 are correlated. This is due to the fact that for all three generators, the scale of the flow parameter depends on the diagonal differences of the matrix. For all three generators the flow in the limit of nearly-diagonal matrices of the generators is similar, where we have a convergence $\propto e^{-|\Delta E|\ell}$ for the gpc-generator and $\propto e^{-|\Delta E|^2\ell}$ for the R1- and R2-generator (see section Sec. 3.2). If a matrix has almost degenerate eigenvalues, all three renormalizing generators perform worse than the generator R3.

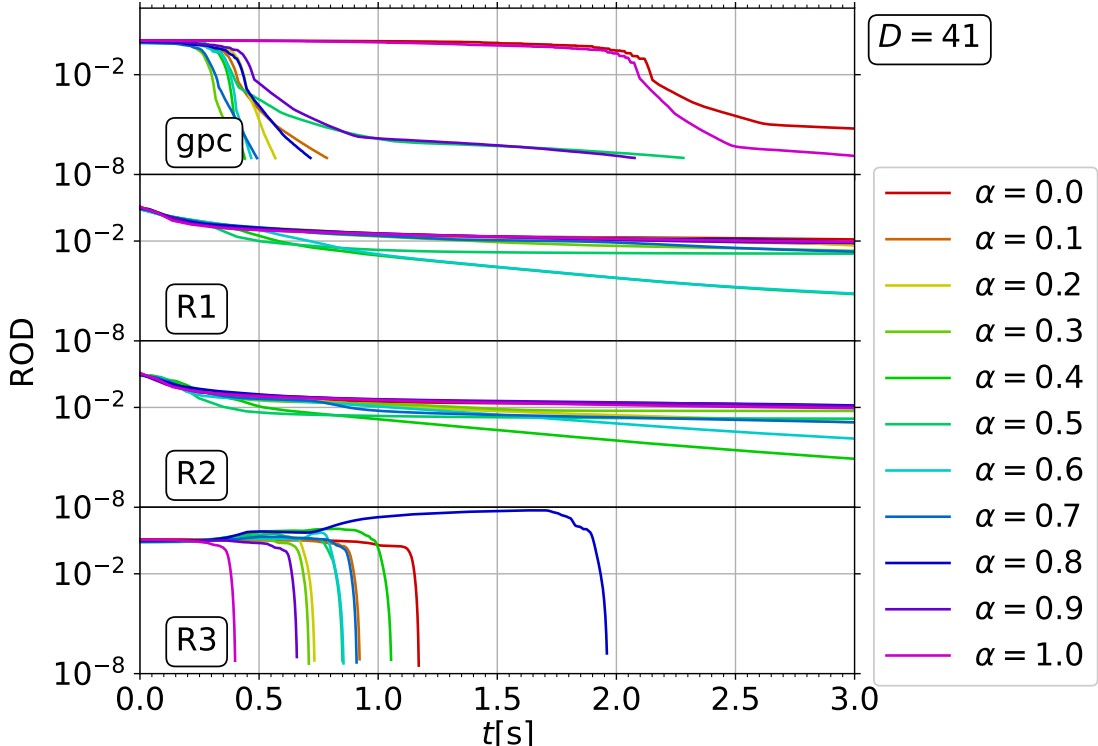

Figure 3: ROD flow vs. time for $D$-dimensional random matrices, see Sec. 4.3.2, and various crossover ratios $\alpha$ induced by the gpc-generator in comparison to flows induced by the R1-, R2- and R3-generator.

Inspecting the panels in Fig. 2 from left to right, the convergence speeds decrease for larger matrices, which is not surprising. For the convergence coefficient $C_{\text{Conv}}^{(t)}$, the calculation of the flow naturally takes longer for larger matrices. For the generator gpc, R1 and R2, the coefficient $C_{\text{Conv}}^{(\ell)}$ decreases with increasing $D$ as well, because the matrices increase in size and the flow becomes more intricate. Furthermore, for larger matrices it becomes more likely for the initial diagonal elements $m_{nn}(0)$ to be close to each other, since these are restricted to the region $[-1, 1]^2 \in \mathbb{C}^2$. This slows down the renormalizing flow at the beginning of the calculation. Note that this effect is only relevant in the initial phase of the calculations, since the eigenvalues, i.e. the final diagonal elements $m_{nn}(\infty)$, scale linearly with $D$ and do not move closer to one another for larger $D$.

We stress that for all plots we checked the actual ROD flow and compared the spectrum

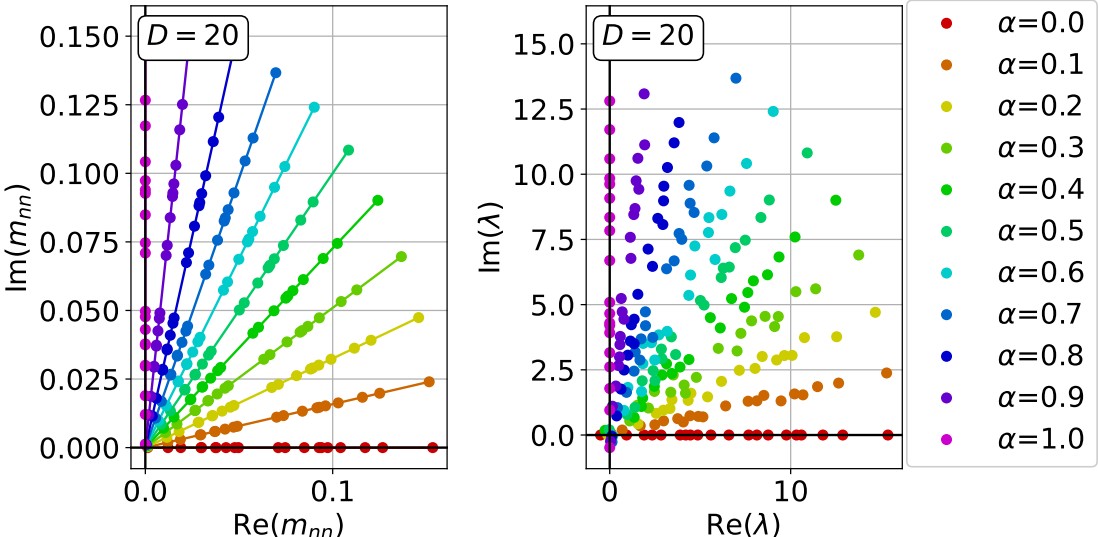

Figure 4: Representative diagonal elements (left) and eigenvalues (right) of the sorted truncated random matrices, see Sec. 4.3.3.

with results obtained by exact diagonalization to ensure that the convergence coefficients correctly reflect the convergence speed. For illustration, we include some exemplary flows without averaging in Fig. 3. For the R3-generator, the convergence coefficient overestimates how quickly the flow converges because of the initial transient, where the step size of the integration is small for R3. But after the slow start a rapid convergence is achieved which exceeds the convergence speed of the gpc-generator. One should note, however, that for the R3-generator the ROD can increase significantly at the beginning. This feature is important in the discussion of the truncation error, see below. Note that no truncation has been performed in the present example so that the error due to truncation does not matter here.

### 4.3.3 Truncation Error

Our tests in the previous section show rapid convergence of the R3-flow, while the renormalizing flows (gpc, R1 and R2) converge more slowly. Since flow equations are used primarily with truncations, fast convergence can be detrimental if it is tied to large truncation errors. Renormalizing flows are generally better at reducing this error. This expection is confirmed in this section.

We benchmark the truncation error using two different models to test two different situations. For the first model we use *unordered truncated random matrices*, which we sample with (62) and introduce an expansion parameter $\lambda$ according to $m_{nj, \text{prep}} = \lambda^{|n-j|} m_{nj}$, explained in Sec. 4.1. This way, the diagonal elements and the spectrum are not ordered, which makes it more difficult for renormalizing flows to keep the error small, see Sec. 2.7.

For the second model we use *ordered truncated random matrices* with ordered diagonal elements. This allows us to check our claim that the renormalizing flow induces smaller truncation errors if the diagonal is ordered. We sample the ordered matrices in the same way as the unordered matrices but finally replace the diagonal elements by

$$m_{00, \text{ordered}} = \exp\left(i\frac{\alpha\pi}{2}\right) r_0 \,, \tag{63a}$$

$$m_{nn, \text{ordered}} = m_{n-1,n-1, \text{ordered}} + \exp\left(i\frac{\alpha\pi}{2}\right) r_n \quad \forall n \in [1, D], \tag{63b}$$

where the $r_n$ are real random numbers drawn from an uniform distribution on $[0, 1]$. The

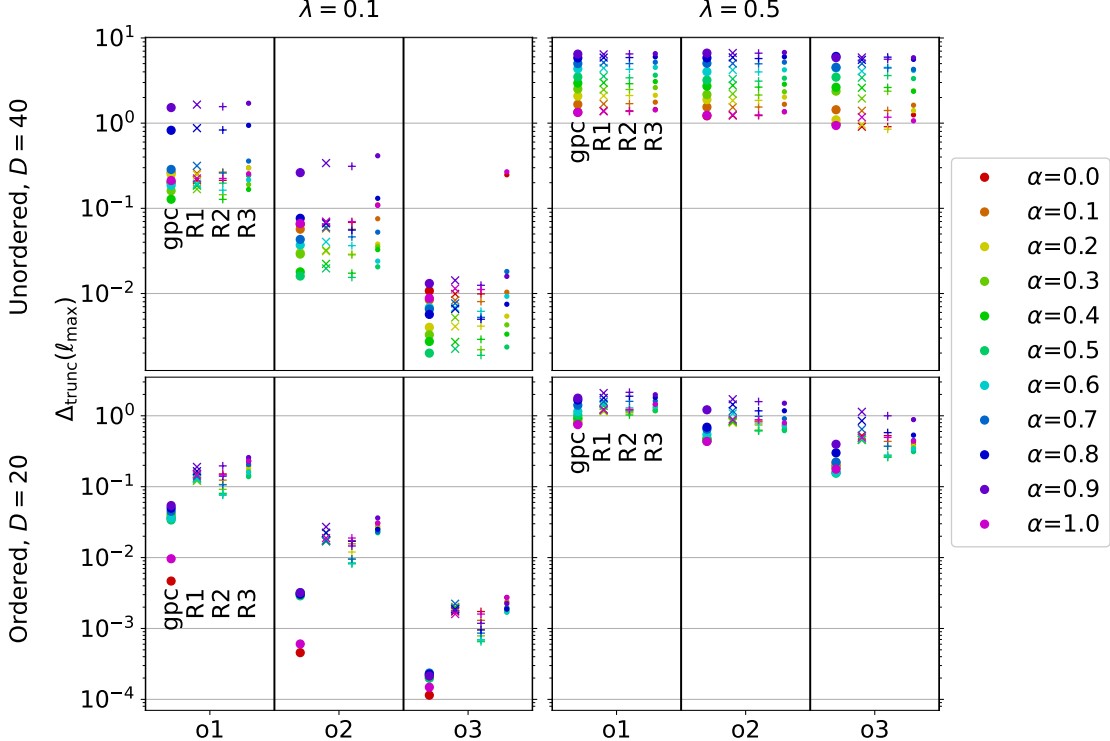

Figure 5: Truncation error $\Delta_{\text{trunc}}(\ell_{\max})$ between exact spectrum of the truncated ordered matrix and the flow equation result of the matrices described in Sec. 4.3.3, averaged over 100 samples each. The top row shows results for random matrices $M$, the bottom row shows results for ordered matrices. The truncation order is denoted by (o1,o2,o3) for truncation in order (1,2,3).

crossover ratio $\alpha$ is the same parameter we used for sampling the random matrix in (62). Note that we do not alter the off-diagonal elements. This way, all diagonal elements lie on a straight line in the complex plane and the distances between the diagonal elements randomly fluctuate between 0 and 1. We show some exemplary ordered diagonal elements in the left panel of Fig. 4 and the resulting spectrum in the right panel of the same figure. We see that both the diagonals as well as the spectrum are ordered nicely. The spectrum does not form a perfect line because of the random off-diagonal elements $m_{nj}$.

The results in truncation order o1, o2 and o3 (referring to 1, 2 and 3 minor diagonals) for the unordered model are shown in the top row of Fig. 5. We see that in this case, all four generators have a similar error $\Delta_{\text{trunc}}$ with the R3-generator performing slightly worse. Note that we average over 100 samples for each data point. For some of the sampled matrices, the gpc-generator has a significantly lower error than the other generators, but the average error does not differ significantly. The bottom row of Fig. 5 shows the results for the ordered model. In this case, we can see clearly that the gpc-generator has the highest accuracy due to its renormalizing property. The R1- and R2- generator are also renormalizing, but perform only slightly better than the R3-generator for a small perturbation parameter $\lambda = 0.1$. Generator R2 is slightly less accurate than R1.

The large error of the R3-generator is the price to pay for the rapid convergence observed in Sec. 4.3.2. The R3-flow treats all matrix elements at the same time, which takes less computation time, but induces significant renormalization of the off-diagonals far away from the diagonal, i.e. the truncated matrix elements. This is in accordance with the initially rising ROD we see in Fig. 3, which reveals that a major reordering happens in the off-diagonal elements.

The renormalizing gpc-generator shows no such problem. It is surprising, however, that R1 and R2 do not perform much better than R3 even though they are renormalizing. This makes R1 and R2 the least well suited generators for this model.

For larger perturbation parameters $\lambda = 0.5$ the differences between the generators become less significant. One reason for this is that the ordering of the diagonal elements can change during the flow. If the off-diagonal elements scaled by $\lambda^{|n-j|}$ are too large, such reordering occurs more frequently. If the ordering is destroyed in this way, the renormalizing property of the gpc-, R1- and R2-generator no longer reduces the error. In fact, the R1 and R2 sometimes perform even worse than R3. Nevertheless, the gpc-generator still shows the highest accuracy even for larger perturbation parameters.

One might be surprised by the fact that the gpc-generator shows a finite error for $\alpha = 0.0$ and $\alpha = 1.0$. Naively, one would expect a vanishing error in these cases, because those edge cases correspond to the pc- or ipc-generator described in Sections 2.2 and 2.6, which conserve band-diagonality and hence no flow should go into the truncated area. However, the diagonal elements can be reordered by the flow, i.e. two diagonal elements cross each other, in which case band-diagonality is not always preserved. For this reason, the gpc-generator has a finite, albeit small, error $\Delta_{\text{trunc}}$ even for $\alpha = 0.0$ and $\alpha = 1.0$.

### 4.4 Ordered Dissipative Scattering Model

### 4.4.1 Physical Model

After discussing a purely mathematical model of random matrices in the previous section, we approach a real physical situation. We consider a model with many fermionic modes, which we solve by integrating the flow equations numerically. Then, we compare the performance of the gpc-flow with the R-flows. We consider a gas of spinless fermions in a $d$-dimensional box of volume $L^d$ with a loss mechanism localized at $\mathbf{x} = 0$ [1]. The Lindblad master equations read

$$i\hbar \frac{\mathrm{d}}{\mathrm{d}t}\rho(t) = [H, \rho(t)] + i\hbar \int \mathrm{d}\mathbf{x}\, \Gamma(\mathbf{x})\left(\Psi(\mathbf{x})\rho(t)\Psi^\dagger(\mathbf{x}) - \frac{1}{2}\left\{\Psi^\dagger(\mathbf{x})\Psi(\mathbf{x}), \rho(t)\right\}\right), \quad (64a)$$

$$\hat{H} = \sum_{\mathbf{k}} \epsilon_{\mathbf{k}} \hat{c}_{\mathbf{k}}^\dagger \hat{c}_{\mathbf{k}}, \quad \Gamma(\mathbf{x}) = \gamma \delta(\mathbf{x}). \quad (64b)$$

In momentum space, the dynamics can be written as

$$i\hbar \frac{\mathrm{d}}{\mathrm{d}t}\rho(t) = \left[\sum_{\mathbf{k}} \epsilon_{\mathbf{k}} \hat{c}_{\mathbf{k}}^\dagger \hat{c}_{\mathbf{k}}, \rho(t)\right] + i\frac{\hbar\gamma}{L^d} \sum_{\mathbf{k},\mathbf{q}}\left(\hat{c}_{\mathbf{k}}\rho(t)\hat{c}_{\mathbf{q}}^\dagger - \frac{1}{2}\left\{\hat{c}_{\mathbf{k}}^\dagger \hat{c}_{\mathbf{q}}, \rho(t)\right\}\right). \quad (65)$$

In the superoperator representation (47) we obtain

$$M = \sum_{\mathbf{k}} \epsilon_{\mathbf{k}}\left(c_{\mathbf{k}}^\dagger c_{\mathbf{k}} + \tilde{c}_{\mathbf{k}}\tilde{c}_{\mathbf{k}}^\dagger\right) - i\frac{\hbar\gamma}{2L^d}\sum_{\mathbf{k},\mathbf{q}}\left(c_{\mathbf{k}}^\dagger c_{\mathbf{q}} - \tilde{c}_{\mathbf{k}}\tilde{c}_{\mathbf{q}}^\dagger\right) - \frac{\hbar\gamma}{L^d}\sum_{\mathbf{k},\mathbf{q}}\tilde{c}_{\mathbf{k}}c_{\mathbf{q}} - \sum_{\mathbf{k}}\left(\epsilon(\mathbf{k}) + i\frac{\hbar\gamma}{2L^d}\right), \quad (66)$$

which reads in matrix form

$$M = \begin{pmatrix} H - \frac{i\hbar}{2}\Lambda_1 & 0 \\ -\Lambda_1 & H + \frac{i\hbar}{2}\Lambda_1 \end{pmatrix}, \quad (67a)$$

$$h_{nj} = \epsilon(\mathbf{k}_n)\delta_{nj}, \qquad (\Lambda_1)_{nj} = \frac{\hbar\gamma}{L^d} \,\forall n, j. \quad (67b)$$

Note that the elements of the matrix are the coefficients of the bilinear operator pairs, so that the computation is performed in second quantization. The triangular block form allows us to

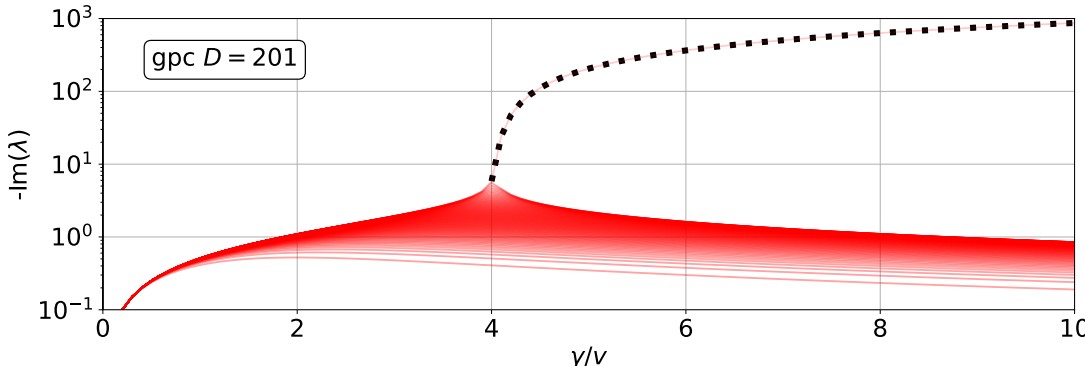

Figure 6: Eigenvalues of the ordered dissipative scattering model, see (68), calculated with the gpc-generator and dimension $D = 201$, i.e. for N=100. The eigenvalues are represented by transparent lines, and the eigenvalue of the strongly dissipative state starting at $\gamma/\nu = 4$ is additionally marked by a dotted black line.

focus on finding the eigenvalues of only the first block. We consider the simplified situation of a one-dimensional linear dispersion $\epsilon(k)$

$$M' = H - \frac{i\hbar}{2}\Lambda_1\,, \tag{68a}$$

$$m_{nj} = \epsilon(k_n)\delta_{nj} - i\frac{\hbar^2\gamma}{2L}\,, \qquad \epsilon(k_n) = \hbar\nu\frac{2\pi}{L}n, \qquad n \in [-N, -N+1, ..., N-1, N]\,. \tag{68b}$$

Note that the matrix dimension is $D = 2N + 1$ with an energy cutoff $|\epsilon| \leq \Lambda_N = \hbar\nu\frac{2\pi}{L}N$. Analytical properties are discussed in App. D. For $\gamma > 4\nu$ one eigenvalue of particular interest

$$\lambda_{\text{sds}} = -i\Lambda_N \tan\left(\frac{\pi}{2}\left(\frac{4\nu}{\gamma} - 1\right)\right), \quad \gamma > 4\nu\,, \tag{69}$$

appears, which has no real part and a much larger imaginary part than the other eigenvalues, corresponding to a *strongly dissipative state* denoted by the subscript 'sds'. This state has a large negative imaginary part of the eigenvalue corresponding to an especially quick decay.

Previous work [1] calculated the flow equations analytically in second quantization, while for our benchmark we calculate them numerically by commuting the matrix representations. To show that this leads to exactly the same flow equations, we carry out the commutation also analytically in App. E.

### 4.4.2 Convergence Speed

For the benchmark of convergence speeds in this model, we only consider the two most efficient generators, gpc and R3, since the other two generators induce a much slower convergence. The resulting imaginary parts of all eigenvalues are shown in Fig. 6 and agree perfectly with prior findings [1]. We can clearly see the strongly dissipative state emerging for $\gamma > 4\nu$.

The convergence coefficient does not represent the ROD flow reliably for this model. If we look at the definition of $\ell_{\max}$ and $\ell_{\min}$ in Sec. 4.1, we see that we defined them so that an initial transient at the beginning of the flow is suppressed. This is correct for calculating the coefficient in $\ell$-space, where the transient is confined to the vicinity of $\ell \approx 0$. If, however, the initial changes of the flow are so big that the step size must be reduced in the integration algorithm so that integrating the initial flow takes a significant portion of the real runtime.

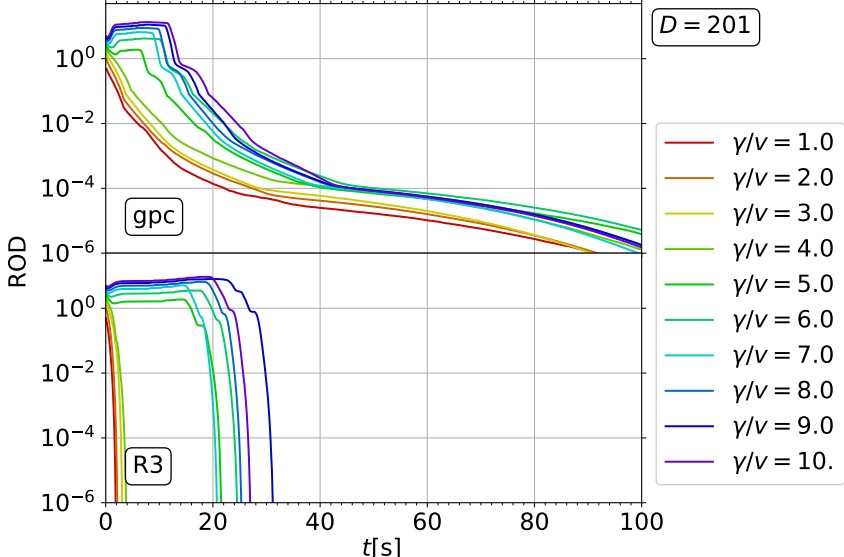

Figure 7: ROD flow of the ordered dissipative scattering model, see (68), calculated with the gpc-generator and R3-generator and $D = 201$.

Hence, the convergence coefficient in real time does not display the convergence behavior of R3 appropriately. In Fig. 7 we see that this problem occurs for the R3-generator for $\gamma > 4v$, where the strongly dissipative state appears. Both the gpc-generator and the R3-generator initially cause an increasing ROD when the strongly dissipative state is present. For the R3-generator, this initial phase takes a significantly higher computation time, slowing it down considerably compared to the cases $\gamma \leq 4v$. However, the R3-generator still converges much more quickly to extremely small RODs than the gpc-generator due to the rapid convergence after the initial transient.

### 4.4.3 Truncation Error

Since this model is a more physical example than the random matrices in Sec. 4.3, it can provide insight in the accuracy of the flows in real applications. To this end, we calculate the truncation error of all generators. We do not sort the diagonal elements for this benchmark,

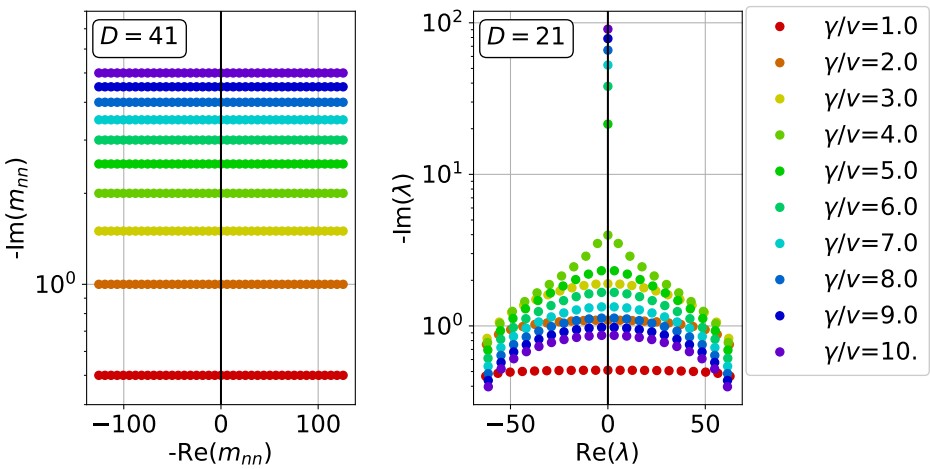

Figure 8: Representative diagonal elements (left) and spectra (right) of the ordered dissipative scattering model, see (68).

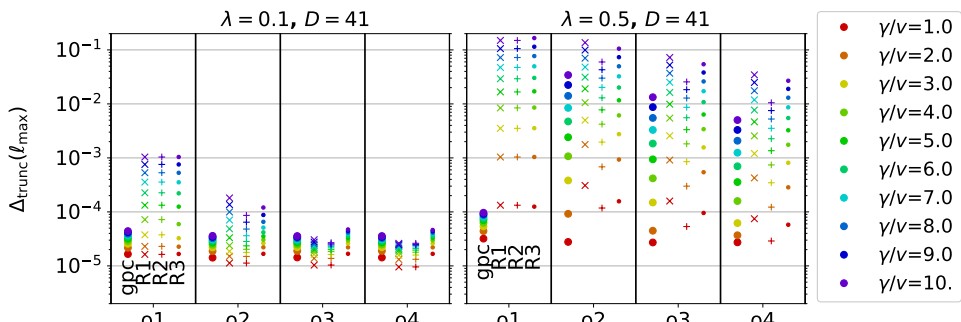

Figure 9: Truncation error $\Delta_{\text{trunc}}(\ell_{\max})$ between exact spectrum of the truncated matrix and the flow equation result for the ordered dissipative scattering model, see (68). The truncation order is denoted by (o1,o2,o3,o4) for truncation in order (1,2,3,4).

because they are already ordered in a nice fashion in the definition of the model (68). This can be seen in Fig. 8, where the diagonal elements are displayed in the left panel and the spectrum is displayed in the right panel. Note that the spectrum is shown for the full system without truncation. We see the single strongly dissipative state emerging as a separated eigenvalue, while all other eigenvalues are ordered on a one-dimensional curve. Such an ordering of the diagonal elements is advantageous for the application of the renormalizing generators gpc, R1 and R2.

In order to study a perturbative truncation we introduce the expansion parameter $\lambda$ as before $m_{nj,\text{prep}} = \lambda^{|n-j|} m_{nj}$ so that off-diagonal elements further away from the diagonal are less relevant. The truncation errors are shown in Fig. 9. We see that the gpc-generator performs better than the other generators most of the time, especially for low truncation orders. This is especially noteworthy because Fig. 7 shows that both the gpc- and the R3-generator initially see an increasing ROD. The gpc-generator, however, mainly changes the matrix elements close to the diagonal which are not truncated while the R3-generator changes the matrix elements far away from the diagonal as well causing a larger truncation error $\Delta_{\text{trunc}}$. This can be seen nicely since $\Delta_{\text{trunc}}$ for gpc and R3 differ the most in truncation order o1 for high values of $\gamma$. High values of $\gamma$ correspond to a dominant strongly dissipative state. This is in accordance with the fact that we see a rising ROD in Fig. 7 when the strongly dissipative state emerges. The performance of R2 and R3 fluctuates: in many cases they perform just as badly as R3, but sometimes they provide even more accurate results than gpc. It is noteworthy, however, that the gpc-generator shows a higher accuracy than R1 and R2 in many cases. This observation agrees with the properties summarized in Tab. 1 which suggested that the accuracy of the gpc-generator is placed between R2 and R3. One might be surprised by the fact that the truncation error $\Delta_{\text{trunc}}$ of the gpc-generator increases for higher orders at $\lambda = 0.5$. As discussed in Sec. 4.1, this is due to the initial matrix $M_{\text{trunc}}$ having more elements in higher truncation orders. The additional matrix elements $m_{nj} = -\mathrm{i}\lambda^{|n-j|}\hbar^2\gamma/(2L)$ are significant for $\lambda = 0.5$, so that a higher truncation order can lead to a more involved flow implying a larger $\Delta_{\text{trunc}}$. This effect does not occur in a real application where the full initial matrix $M_{\text{trunc}}(0)$ is already captured by a low truncation order so that increasing the truncation order will decrease the truncation error $\Delta_{\text{trunc}}$.

## 4.5 Disordered Dissipative Scattering Model

### 4.5.1 Physical Model

While the last example featured ordering in the form of a linear, one-dimensional dispersion we want to pass on to a model closer to a real physical situation. We do so by introducing disorder while still focusing on a system with a single strongly dissipative state [1]. In this way, we can compare the performance of the generators for a real physical system with suboptimal conditions for the renormalizing generators. We use a fermionic disordered tight-binding model on a one-dimensional chain

$$\hat{H} = -J \sum_j \left[ \hat{c}_j^\dagger \hat{c}_{j-1} + \text{h.c.} \right] + \sum_j h_j \hat{n}_j \quad h_j = \text{random}([-W, W]), \tag{70}$$

with periodic boundaries and $h_j$ drawn from a uniform distribution, where $j$ denotes the lattice sites. By adding a localized loss rate $\gamma$ at site 0 we obtain the Lindblad master equations

$$i\hbar \frac{d}{dt} \hat{\rho}(t) = [\hat{H}, \hat{\rho}(t)] + i\hbar\gamma \left( \hat{c}_0 \rho(t) \hat{c}_0^\dagger - \frac{1}{2} \{ \hat{n}_0, \rho(t) \} \right). \tag{71}$$

Similar to the ordered model in Sec. 4.4.1, we use fermionic superoperators to obtain the matrix form (47) of the master equations with the elements of matrix $M$

$$m_{nn}(\ell = 0) = h_n - i\frac{\hbar\gamma}{2} \delta_{n,0}, \tag{72a}$$

$$m_{nj}(\ell = 0) = -J(\delta_{n,j+1} + \delta_{n,j-1}). \tag{72b}$$

From $m_{nn}(\ell = 0)$ we can expect already that a large $\gamma$ leads to the emergence of a single strongly dissipative state with a large negative imaginary value. We checked by exact diagonalization that we obtain the same results as in Ref [1]. In particular, we checked that a strongly dissipative state emerges for $\gamma \geq 4J$.

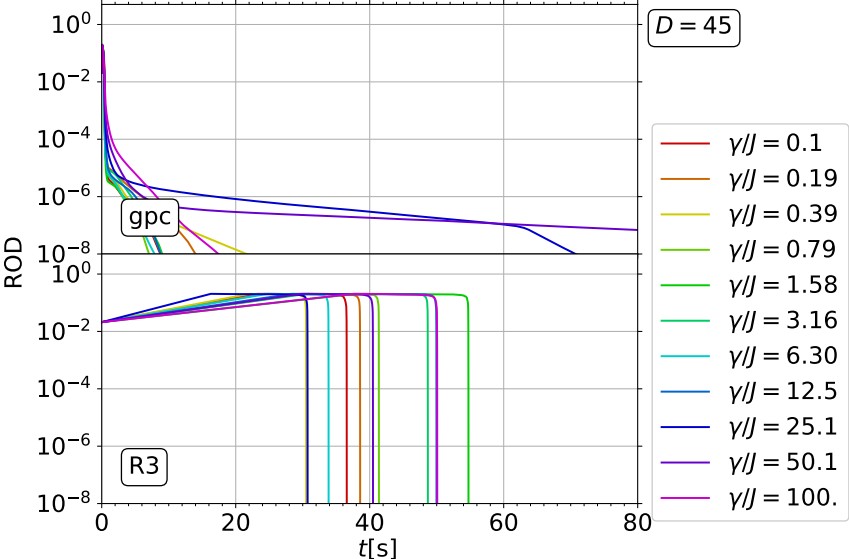

Figure 10: Exemplary ROD flow of the disordered dissipative scattering model, defined in (72), for $N = 45$ and $W = J$, calculated with the gpc-generator and R3-generator, averaged over 10 samples.

### 4.5.2 Convergence Speed

Fig. 10 depicts exemplary flows of the ROD for gpc and R3, averaged over 10 samples. We see that the R3-flow again takes a long time with an initially rising ROD, so the convergence coefficients are an inadequate benchmark measure. Opposed to the benchmark results for the ordered dissipative scattering model in Fig. 7, the ROD of the R3-generator now increases even for $\gamma < 4J$, where no dominant strongly dissipative state appears. Furthermore, the gpc-generator does not suffer from an increasing ROD in any calculation and in most cases it even beats R3 in pure convergence speed. One possible explanation is the fact that in the disordered dissipative scattering model the differences between the diagonal elements are bigger than in the ordered model, leading to a faster gpc-convergence $\exp(-|\Delta E| \ell)$.

### 4.5.3 Truncation Error

We benchmark the truncation error on the unaltered system, which means without ordering the diagonal elements $m_{nn}$ and without introducing an expansion parameter $\lambda$. The scaling is not necessary since the matrix is essentially already tridiagonal aside from the two furthest off-diagonal elements $m_{0,D-1} = m_{D-1,0} = -J$, which are always truncated in our truncation scheme. Therefore, all truncation orders use the same initial truncated matrix and increasing the order typically increases the accuracy. We do not order the diagonal elements since this would break the tridiagonal form of $M_{\mathrm{trunc}}$, which would cause initial off-diagonal elements $m_{nj}(0)$ to be truncated.

The truncation error $\Delta_{\mathrm{trunc}}$ is shown in Fig. 11. We find that $\Delta_{\mathrm{trunc}} \in [10^{-1}, 10^1]$ is quite large for all generators. This is explained by the disordered energies, which force even the three renormalizing generators (gpc, R1 and R2) to perform significant renormalizations of the off-diagonal elements far away from the diagonal, which are truncated. In this case, high accuracy requires a high truncation order. We see that the gpc-generator performs with the highest accuracy of all generators in most cases. The difference between different generators is greater when the range $[-W, W]$ of the randomly sampled energies $h_n$ is large. The R1- and R2- generator also perform quite well with R3 generating the largest truncation error. This is surprising, since in Sec. 4.3.3 we have seen for random matrices that the renormalizing generators perform with high accuracy only if the diagonal matrix elements are ordered. Here we see that they perform well even in presence of disorder.

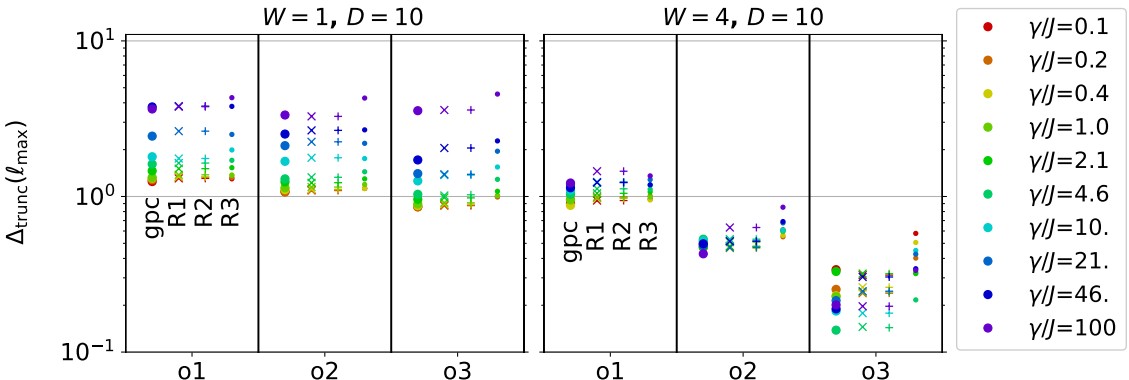

Figure 11: Deviation $\Delta_{\mathrm{trunc}}(\ell_{\max})$ between exact spectrum of the truncated matrix and the flow equation result for the disordered dissipative scattering model (72), averaged over 100 samples. The truncation order is denoted by (o1,o2,o3) for truncation in order (1,2,3).

## 4.6 Random Lindbladians

### 4.6.1 Matrix Generation

In Sec. 4.3, we sampled random matrices. Subsequently, we considered two selected physical models, one with ordered energies and the other with disordered energies. In this section, we approach a wider range of large open quantum systems which may represent real applications of dissipative flow equations better. To achieve this, we sample random Lindbladians because the Lindblad master equations in Sec. 2.4 are the most general Markovian description of open quantum system. The sampling of Lindbladians is still a subject of current studies [73].

Fig. 12 shows the difference between the spectrum of the uniformly random complex elements used for the first test of the gpc-generator with random matrices in Sec. 4.3 and the spectrum of the random Lindbladians considered here. While a random matrix has a circular spectrum centered at the origin, a random Lindbladian has one eigenvalue 0 and a cluster of eigenvalues centered around -i in a lemon-shape [74]. Note that the circular shaped spectrum of the random matrix does not contradict the spectrum shown in Fig. 4 in Sec. 4.3.3, because that figure displayed the spectrum of the artificially sorted matrix for the truncation benchmarking, which differs from the purely random matrix in (62). Compared to the examples in Sections 4.4 and 4.5, where we had a single strongly dissipative state with a dominating imaginary part, the random Lindbladians in this section show a cluster of approximately equally dissipative states and the obligatory stationary state.

To sample the Lindbladians, we use the Gorini-Kossakowski-Sudarshan-Lindblad form [75, 76]

$$\mathcal{L}(\rho) = [H, \rho] + \mathcal{L}_D(\rho) = \mathcal{L}_U(\rho) + \mathcal{L}_D(\rho), \tag{73}$$

where $\mathcal{L}_D(\rho)$ denotes the dissipative part

$$\mathcal{L}_D(\rho) = i\hbar \sum_{m,n=1}^{N^2-1} K_{mn} \left[ F_n \rho F_m^\dagger - \frac{1}{2} \left( F_m^\dagger F_n \rho + \rho F_m^\dagger F_n \right) \right], \tag{74}$$

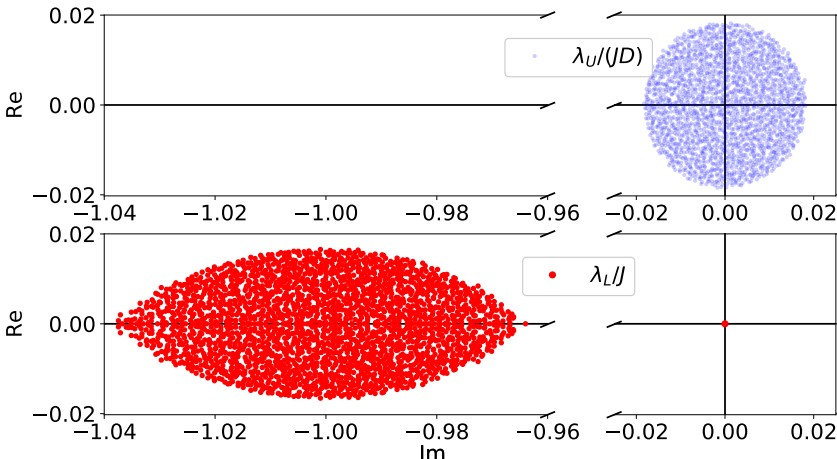

Figure 12: Top: Circular spectrum $\{\lambda_U\}$ of a matrix with uniformly random complex elements sampled with (62) for $\alpha = 0.5$ and dimension $D = 2000$. The spectrum is normalized to the matrix dimension $D$ and energy scale $J$. Bottom: Spectrum $\{\lambda_L\}$ of a random Lindbladian, see Sec. 4.6.1, normalized to the energy scale $J$. The Lindbladian is sampled with $N = 50$ states, yielding $D = N^2 = 2500$ eigenvalues. The spectrum consists of a single stationary state $\lambda_L = 0$ and a lemon-shaped cluster around $-i$.

with an orthonormal Hilbert-Schmidt basis $\{F_n\}$, $n = 1, 2, ..., N^2 - 1$ of traceless matrices in Fock-Liouville space with $\text{Tr}(F_n) = 0$, $\text{Tr}(F_n F_m^\dagger) = \delta_{n,m}$ and a positive semidefinite, complex Kossakowski matrix $K$ sampled from complex square Wishart matrices $W = GG^\dagger \geq 0$ with a complex square Ginibre matrix $G$ with independent complex Gaussian elements [74]. The particular choice for sampling $K$ does not matter, since the spectral features of the random purely dissipative Lindbladians are universal. To set up the Hilbert-Schmidt basis, we use the $N^2 - 1$ Hermitian generators of SU($N$) [77]. The sampling process and the creation of the matrix representation are explained in more detail in App. F.

### 4.6.2 Convergence Speed

Fig. 13 shows the ROD-flow of the generators for random Lindbladians (74) for two different dimensions. Like for the other examples, the R1- and R2-generator perform very poorly, taking a very long time to decrease the ROD by only a few orders of magnitude. This is not too surprising, since convergence of these generators scales with $|\Delta E|^2$ and the differences $\Delta E$ inside the cluster of eigenvalues become very small. The gpc-generator with its $|\Delta E|$-convergence performs better. One would expect the R3-generator to converge even more rapidly, because it does not depend on energy differences. For the smaller system with $D = 100$, this is true: After an initially rising ROD, the R3-generator shows rapid convergence, like it does for most of the previously considered models. For larger systems such as $D = 400$, however, the R3-generator shows no rapid convergence and actually does not converge at all in our calculation which continues up to $t = 150000\,\text{s}$ (not shown).

It is also striking that the ROD increases much more strongly for the R3-generator than for the gpc-generator for all system sizes. A possible explanation is the denominator $m_{nn} - m_{jj}$ in the R3-generator (26c), which might cause numerical instabilities. In the numerical integration we try to avoid this by setting $\eta_{nj}^{\text{R3}} = 0$ if $|m_{nn} - m_{jj}| < 10^{-10}$, but this numerical cutoff can also create numerical instabilities if two diagonal matrix elements are close to each other but their overlapping off-diagonal $m_{nj}$ has not been eliminated. This unstable behavior is an important caveat of the R3-generator which does not arise for the more robust gpc-flow. In the other models studied, we did not find unstable flows for the R3-generator. The fragility of the R3-flow for large Lindbladians puts the rapid convergence observed in most other cases into perspective. The gpc-generator converges less rapidly in most cases, but more reliably.

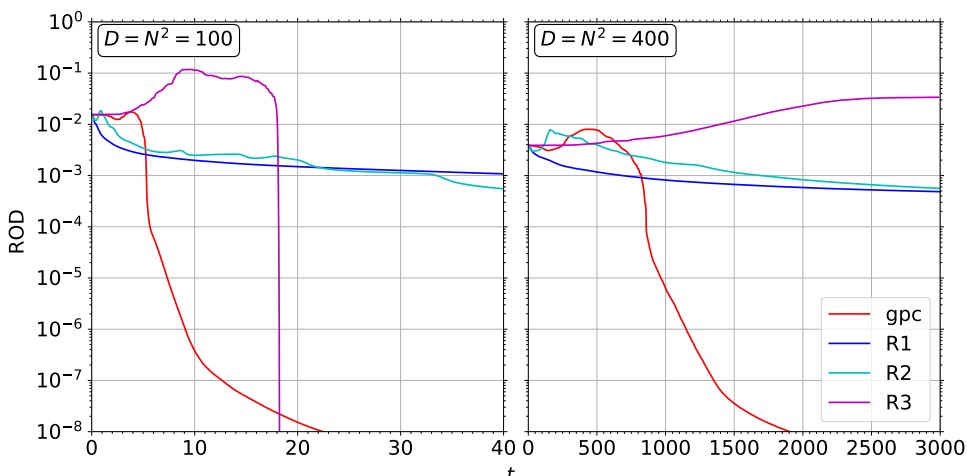

Figure 13: ROD flow of random Lindbladians, defined in Sec. 4.6, calculated with the gpc-generator, R1-, R2- and R3-generator, for a Lindbladian with 100 ($N = 10$) and 400 ($N = 20$) eigenvalues.

# 5 Conclusions

Open quantum systems in general and Lindblad master equations in particular describe quantum systems in interaction with an external bath. They are important for the theoretical description of novel physics such as non-equilibrium physics, relevant for all pump-probe setups, and quantum information processing. Lindbladian systems are described by non-Hermitian matrices which calls for generalizations of existing methods such as the flow equation scheme. Flow equations are a powerful renormalization tool and their renormalizing flow is calculated by choosing an appropriate generator scheme. The dependence on the energy difference is central for the renormalizing property of the generators and the convergence speed. A robust, renormalizing flow often converges more slowly because not all matrix elements are treated at the same time and speed. But such a flow is less prone to truncation errors. Therefore, one must typically compromise between rapid convergence and minimal truncation error. This aspect has been elucidated in the present manuscript and represents its main punchline compared to previous studies.

We introduced the gpc-generator, a generalization of the pc-generator to non-Hermitian matrices, and calculated the general form of its flow equations. In the limiting cases of Hermitian and Antihermitian matrices, the gpc-generator is equivalent to the pc- or ipc-generator. We presented a proof for convergence and showed that band-diagonality is preserved only in very limiting cases, namely for complex-sorted matrices. We compared the flow equations with the R-generators from Ref. [1] and introduced the generalization $\eta^{(r)}$ in Sec. 3.2. The generator $\eta^{(1)} = \eta^{R2}$ has a quadratic energy dependence and sacrifices convergence speed for higher accuracy and the non-renormalizing generator $\eta^{(-1)} = \eta^{R3}$ sacrifices accuracy for rapid convergence by treating all matrix elements simultaneously. The linearly energy-dependent $\eta^{(0)} = \eta^{gpc}$ is placed between those two.

In numerical tests, we found that the gpc-generator converges much more quickly than the R1- and R2-generators in all considered cases. The R3-generator, which does not depend on energy differences, typically converges more quickly than the other generators down to the typical ROD values of ROD$< 10^{-8}$ where the matrix is numerically diagonal. This result coincides with the expectations based on the energy scaling. Surprisingly, for some limiting cases of randomly sampled Lindbladians, the gpc-flow shows a more robust and fast convergence than the R3-flow. A possible explanation is that the R3-flow becomes numerically unstable when a matrix has almost degenerate eigenvalues. We stress that our analysis is based on real computation time and not on the convergence behavior over the flow parameter $\ell$, which cannot be compared properly due to the different units of $\ell$ for different generators.

We cannot overemphasize that truncations are the main issue in the applications of flow equations because simple systems which are tractable without any truncation can generically also be solved by other methods. Therefore, the truncation error $\Delta_{trunc}$ has to be minimized so that the relevant physics is captured well. The fast R3-generator suffers from large truncation errors between the exact spectrum and the spectrum calculated using the flow equations. The R1- and R2- generators induce smaller errors in many cases, especially when the diagonal elements are ordered. In this case, the renormalizing property of these generators causes less flow into matrix elements which are truncated. The gpc-generator shows a significantly higher accuracy than the R3-generator in even more cases than R1 and R2 do. Suprisingly, the gpc results are often more accurate than the results for R1 and R2. In the rare cases where gpc is slightly less accurate, it is still worth choosing gpc over R1/R2 since the faster convergence speed means that higher truncation orders are computationally feasible. This also contributes to increasing the overall accuracy of the results.

The gpc-generator $\eta^{(0)} = \eta^{gpc}$ is chosen to fill the gap between the other generators, so that it induces faster convergence than the slow, but accurate $\eta^{(1)} = \eta^{R2}$ with smaller truncation-

Table 2: Results of the analytical analysis and numerical benchmark for the generators considered in this work. The dimension of $\eta$ and $\ell$ are given in terms of the energy $E$ and the asymptotic convergence in terms of the energy differences $\Delta E$ of the system. The convergence speed and the accuracy (during truncation) are depicted by $+$ for positive results, $-$ for negative results and $\sim$ for mixed results.

| Generator | $[\eta]$ | $[\ell]$ | Asymptotic convergence | Convergence Speed | Accuracy |
|---|---|---|---|---|---|
| R1, R2 | $E^2$ | $1/E^2$ | $\exp(-\ell\lvert\Delta E\rvert^2)$ | $-$ | $\sim$ |
| gpc | $E$ | $1/E$ | $\exp(-\ell\lvert\Delta E\rvert)$ | $+$ | $+$ |
| R3 | $1$ | $1$ | $\exp(-\ell)$ | $++$ | $-$ |

errors than the fast, but prone to truncation errors $\eta^{(-1)} = \eta^{\text{R3}}$. Surprisingly, our results show that the gpc-generator $\eta^{(0)} = \eta^{\text{gpc}}$ also provides higher accuracy than $\eta^{(1)} = \eta^{\text{R2}}$ and in some cases converges more quickly than $\eta^{(-1)} = \eta^{\text{R3}}$. These qualitative results are summarized in Tab. 2.

We conclude that the gpc-generator and the R3-generator both present strong tools for studying dissipative systems and Lindbladian master equations. The R3-generator promises fast convergence, while the gpc-generator represents a more consistent renormalization tool with high accuracy in spite of truncations. Just like the pc-generator has seen further development with novel truncation and approximation schemes [49, 66], further studies are called to optimize the gpc- and R3-generator for more complicated systems with sophisticated interactions and dissipation at play. Additionally, generators $\eta^{(r)}$ with $-1 < r < 0$ are promising candidates for further studies and might improve on the convergence speed of the gpc-generator while maintaining high accuracy.

# Acknowledgements

**Author contributions**   G.S. performed all calculations and wrote the manuscript. G.S.U. designed and supervised the project and edited the manuscript.

**Funding information**   G.S.U. is supported by the Deutsche Forschungsgemeinschaft (DFG, German Research Foundation) in the frame of the International Collaborative Research Centre TRR 160 also supported by the Russian Foundation for Basic Research. G.S. and G.S.U. are both also funded by the DFG through Grant No. UH90/13-1.

# A   Other Considered Generator Schemes for Non-Hermitian Matrices

While attempting to generalize the gpc-generator, we tested other generalization schemes which proved less useful for real applications. We present those here to inform the reader about less promising trials.

## A.1 Phase-Shifted PC-Generator

### A.1.1 Definition

A possible generalization of the pc-generator and the ipc-generator for general matrices $M$ is the *phase-shifted particle conserving generator (ppc-generator)*

$$\eta_{nj}^{\text{ppc}}(\theta)[M] = \text{sign}(n-j)e^{i\theta} m_{nj}, \qquad \theta \in \left[0, \frac{\pi}{2}\right]. \tag{75}$$

The idea is that for purely Antihermitian matrices, we can use the ipc-generator, which is just the pc-generator shifted by a phase factor $\exp(i\frac{\pi}{2})$. Randomly drawn matrices that are mostly Hermitian converge nicely for the pc-generator and matrices that are mostly Antihermitian converge well for the ipc-generator. For matrices consisting of both Hermitian and Antihermitian parts, changing the phase shift might lead to a better convergence.

One caveat is the fact that for a given problem a suitable $\theta$ must be chosen. For convenience sake, one can use $\theta = \frac{\pi}{4}$ for all matrices, such that the Hermitian and Antihermitian parts both converge. It should be assumed, however, that choosing $\theta$ in accordance with the ratio of the Hermitian and Antihermitian parts of $M$ leads to a faster convergence, which will be explored below.

### A.1.2 Convergence Speed

The ppc-generator was tested with different phases for different randomly generated $(DxD)$-matrices. Two parameters are important and are varied during the tests

$$\text{phase ratio } \varphi \qquad \theta =: \frac{\pi}{2}\varphi, \tag{76a}$$

$$\text{crossover ratio } \alpha \qquad M := (1-\alpha)(R+R^\dagger) + \alpha(R-R^\dagger) = R + (1-2\alpha)R^\dagger. \tag{76b}$$

The phase ratio $\varphi$ is used instead of $\theta$ for convenience sake. For $\varphi = 0$ the ppc-generator is equivalent to the pc-generator, for $\varphi = 1$ the ppc-generator is equivalent to the ipc-generator.

The crossover ratio $\alpha$ is used for constructing the initial matrix $M$ that is solved via the flow equations. It is constructed from a random complex matrix $R$ such that for $\alpha = 0$ the matrix $M$ is Hermitian and for $\alpha = 1$ it is Antihermitian. In a real application, where the matrix is already given and $\alpha$ is unknown, a reasonable value of $\alpha$ can be obtained by computing

$$\alpha = \frac{\|M - M^\dagger\| - \|M + M^\dagger\|}{2\|M\|}, \tag{77}$$

with a norm that ensures $\alpha \in [0, 1]$, so that one can choose $\varphi = \alpha$ such that the matrix converges nicely.

Fig. 14 shows the ROD-flow and the convergence coefficients for various $\varphi$ and $\alpha$. Recall that a larger coefficient is desirable as it stands for faster convergence. For this comparison, we use $\text{ROD}(\ell)$ instead of $\text{ROD}(t)$ since changing the phase ratio $\varphi$ does not change the energy scale of $\ell$. It can be seen that the fastest convergence is achieved for fully Hermitian matrices $\alpha = 0$ (top right panel) with the pc-generator ($\varphi = 0$) and for fully Antihermitian matrices $\alpha = 1$ (bottom right panel) with the ipc-generator ($\varphi = 1$). Convergence is still fast for $\varphi \approx \alpha$, but no convergence can be achieved for orthogonal angles $|\varphi - \alpha| = 1$ and even $|\varphi - \alpha| \approx 1$ converges very slowly. This means that the pc-generator does not converge to a diagonal matrix for an Antihermitian matrix and the ipc-generator does not converge for a Hermitian matrix.

Convergence is always comparatively slow for $\alpha \approx 0.5$ (bottom left panel), no matter which $\varphi$ is chosen for the generator. However, in those cases the fastest convergence is often

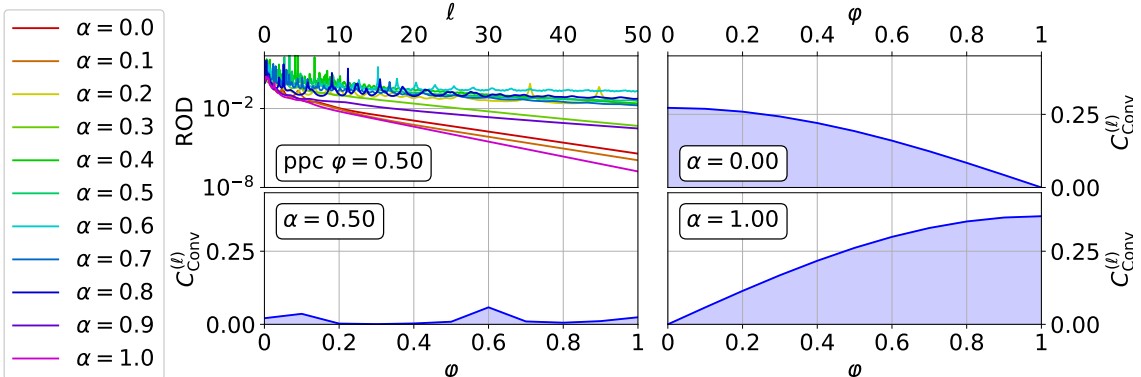

Figure 14: Top left: Exemplary ROD flow of the ppc-generator, see (75), with fixed $\varphi = 0.5$ for mixed random matrices with varying $\alpha$. The other three panels show the convergence coefficients, each for a fixed random matrix with given $\alpha$, diagonalized by the ppc-generator with various values of $\varphi$. All matrices are of dimension $D = 20$.

achieved by a generator with $\varphi \notin \{0, 1\}$. Therefore, the ppc-generator does offer a benefit over the pc- or ipc-generator in those cases. Fastest convergence, however, is not always achieved by $\varphi = \alpha$. The choice of $\varphi$ for optimum convergence speed is not trivial.

Instead of searching an optimal value of $\varphi$, one can use $\varphi = 0.5$ as a standard value to ensure convergence for any complex matrix. For $\varphi = 0.5$, we see convergence of ROD($\ell$) for all $\alpha = [0, 1]$ (top left panel). However, for $\alpha \approx 0.5$ convergence is very slow and the ROD increases periodically. Note that we see no such effect for the gpc-generator in Fig. 3 and that the gpc-generator also displays higher convergence coefficients at $\alpha \approx 0.5$ in Fig 2.

We conclude that the ppc-generator with $\varphi \in [0, 1]$ has the benefit of converging for more matrices than either the pc-generator or the ipc-generator, respectively. However, the ppc-generator has two important drawbacks:

1. It is not obvious which $\varphi$ maximizes the convergence speed for a given matrix. Sometimes, the more trivial pc-generator or the ipc-generator lead to a faster convergence.

2. The ppc-generator converges slower than the gpc-generator and suffers from periodically increasing RODs.

## A.2 Hermitized PC-Generator

### A.2.1 Definition

If the matrix $M$ is not Hermitian, then a Hermitian operator

$$H := M^\dagger M, \qquad h_{nj} = \sum_k m_{kn}^* m_{kj}, \tag{78}$$

can be used as the Hamiltonian while $M$ is treated like an observable, leading to the *hermitized-particle-conserving generator (hpc-generator)*

$$\eta_{nj}^{\mathrm{hpc}}[M] := \eta_{nj}^{\mathrm{pc}}[H] = \mathrm{sign}(n-j) \sum_k m_{kn}^* m_{kj}. \tag{79}$$

The flow equations can be treated by applying the flow equations on $M$ and $H$ at the same time or alternatively by only applying it to $H$ and calculating $\eta$ directly from $H$.

### A.2.2 Fixed Points of the PC- and HPC-Generator

The fixed points of the pc-generator $\eta^{\text{pc}}[H]$ for Hermitian matrices are diagonal matrices. This is easy to show by using the proof that Mielke used for the pc-generator [50]. We can see that if the flow for the diagonal elements has stopped

$$\partial_\ell h_{nn} = 0 \tag{80a}$$

$$\Rightarrow \sum_{k>n} |h_{nk}| = 0 \tag{80b}$$

$$\Rightarrow |h_{nj}| = 0 \quad \forall n \neq j \,, \tag{80c}$$

diagonalization is achieved. Therefore, the fixed points of $\eta^{\text{hpc}}_{nj}[M]$ are all bases in which $H$ is diagonal

$$\forall n \neq j : 0 = h_{nj} \tag{81a}$$

$$= \sum_k m^*_{kn} m_{kj} \tag{81b}$$

$$= \underline{m}^\dagger_j \underline{m}_n \,. \tag{81c}$$

In the last step we see that the condition can be written by using the usual inner product. Considering the case $n = j$ as well, this condition becomes

$$\underline{m}^\dagger_j \underline{m}_n = \delta_{nj} |\vec{m}_n|^2 \,. \tag{82}$$

This resembles the condition of a unitary matrix, where the column vectors of the matrix are pairwise orthonormal with respect to the usual inner product. The column vectors in (82), however, are only pairwise orthogonal.

Thus, the fixed points of the flow, where $M$ converges while $H$ converges to a diagonal matrix, are those where (82) is fulfilled. Diagonal matrices are a subset of such matrices, but convergence to a diagonal matrix $M$ is not guaranteed.

### A.2.3 Convergence Speed

We use the hpc-generator to diagonalize random matrices like we did for the ppc-generator. In the top panel of Fig. 15 we show the flow of the ROD$[M^\dagger M]$ of the Hermitian matrix which converges nicely. This is not surprising, since effectively we are just diagonalizing a Hermitian matrix using the pc-generator. The bottom panel shows the ROD that we care about: ROD$[M]$ for the observable $M$, which is the matrix that we actually want to diagonalize. This ROD$[M]$, however, converges only if $M$ is predominantly Hermitian. In most cases, $M$ retains significant non-diagonal elements, rendering the effort of integrating the flow equations superfluous. Therefore, the hpc-generator offers no advantages when compared to the simple pc-generator approach and actually suffers from the increased numerical effort of solving the flow equations for $H$ and $M$ at the same time.

### A.3 Switching between PC- and IPC-Generator

If the flow equations preserves the Hermiticity or Antihermiticity of the matrices, then the pc- and ipc-generator can be used one after another, i.e.

$$M = H + A := \frac{M + M^\dagger}{2} + \frac{M - M^\dagger}{2} \,, \tag{83a}$$

$$\xrightarrow{\eta^{\text{ipc}}[H]} M' = H' + A'_{\text{diag}} \,, \tag{83b}$$

$$\xrightarrow{\eta^{\text{pc}}[A]} M'' = H''_{\text{diag}} + A''_{\text{diag}} \,, \tag{83c}$$

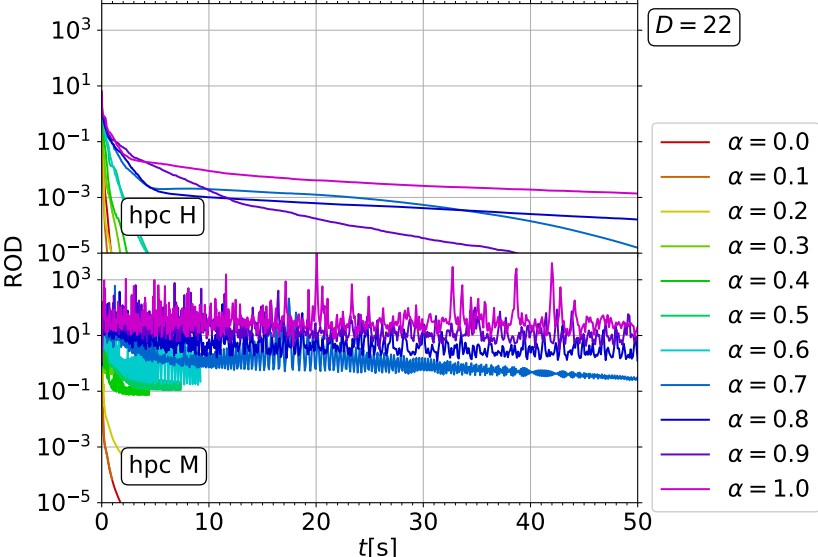

Figure 15: ROD flow of $\text{ROD}[H = M^\dagger M]$ (top) and $\text{ROD}[M]$ (bottom) of the hpc-generator, see (79), for random matrices with different crossover ratios $\alpha$, see Sec. 4.3.

where $H$ and $A$ are always defined by the Hermitian and Antihermitian parts of the respective matrix. The ipc-flow diagonalizes the Antihermitian part $A'_{\text{diag}}$ and then the pc-flow diagonalizes the Hermitian part $H''_{\text{diag}}$. For this approach to diagonalize $M$ completely, the pc-flow must preserve the diagonality of the Antihermitian part so that $A''_{\text{diag}}$ is still diagonal. We show that this naive assumption is wrong by recalling the flow equations of the Antihermitian components (19)

$$\partial_\ell a_{nj} = \text{sign}(n-j)\big[h_{nj}(a_{jj} - a_{nn}) + a_{nj}(h_{jj} - h_{nn})\big] \tag{84a}$$

$$+ \sum_{k \neq n,j}\big[\text{sign}(n-k)\big(h_{nk}a_{kj} + a_{nk}h_{kj}\big) + \text{sign}(j-k)\big(a_{nk}h_{kj} + h_{nk}a_{kj}\big)\big]. \tag{84b}$$

Even tough $A'_{\text{diag}}$ is already diagonal, i.e. $a_{nk} = 0 \;\forall n \neq k$, the flow still yields a finite contribution

$$\partial_\ell a_{nj} = \text{sign}(n-j)h_{nj}(a_{jj} - a_{nn}), \tag{85}$$

introducing off-diagonal elements to $A''$. Therefore our naive assumption that the pc-generator preserves the diagonality of the Antihermitian part is wrong and $A''$ is not guaranteed to be diagonal. If we reverse the order of the generators, starting with pc and following with ipc, the analogue argument holds that the ipc-generator does not preserve the diagonality of the Hermitian part.

Our numerical calculations on random matrices, sampled as defined in Sec. 4.3, confirm that applying the pc-generator and ipc-generator alternately does not lead to any improvement beyond applying either the pc-generator or the ipc-generator once.

# B Loss of Band-Diagonality

One great advantage of the pc-generator is the fact that it preserves the band-diagonality of Hermitian matrices, which is shown in Sec. 2.2. This fact can be used to reduce the number of variables in the flow equations if one uses an operator representation for each matrix

element or a sparse matrix representation. While the pc-generator always preserves the band-diagonality of Hermitian matrices, the gpc-generator does so for a certain class of matrices, but unfortunately not for all complex matrices. We define this class of matrices as

$$M = \{m_{nj}\} \text{ is } complex\text{-}sorted \iff m_{nn} = c x_n, c \in \mathbb{C}, x_n \in \mathbb{R}, x_{j+1} > x_j \forall n, j, \tag{86}$$

which means that

1. all diagonal elements lie on a straight line through the origin in $\mathbb{C}$ and

2. they are sorted on this line.

An example for the first condition are the sorted truncated random matrices introduced in Sec. 4.3.3. The diagonal elements lie exactly on a straight line, which is shown in the left panel of Fig. 4. The first condition is a strong restriction, but if it is fulfilled the second condition can always be met by simply reordering indices. However, reordering indices includes changing the position of off-diagonal elements, which can increase the diagonal width of the matrix, i.e. the number of minor diagonals including the diagonal with non-vanishing elements. Note that matrices with degeneracies in the diagonal elements do not fulfill this condition. Hermitian matrices and Antihermitian matrices automatically fulfill the first condition.

We show

$$\begin{array}{l} \text{If } M(\ell) = \{m_{nj}(\ell)\} \text{ is } complex\text{-}sorted \\ \Rightarrow \text{ gpc-generator preverses band-diagonality of } M \text{ infinitesimally, i.e.} \\ \quad \partial_\ell m_{nj} = 0 \quad \forall |n-j| > \delta \text{ if } M \text{ has diagonal width } \delta \,. \end{array} \tag{87}$$

**Proof:**
A matrix is band-diagonal with diagonal width $\delta$ if

$$m_{nj} = 0 \,\forall |n-j| > \delta \,. \tag{88}$$

We must prove that for all matrices fulfilling (88) $\partial_\ell m_{nj} = 0 \,\forall |n-j| > \delta$ is implied. We first show

$$\frac{m_{nn}^* - m_{kk}^*}{|m_{nn}^* - m_{kk}^*|} + \frac{m_{jj}^* - m_{kk}^*}{|m_{jj}^* - m_{kk}^*|} \overset{(86)}{=} \frac{c^*}{|c|} \left( \frac{x_n - x_k}{|x_n - x_k|} + \frac{x_j - x_k}{|x_j - x_k|} \right) \tag{89a}$$

$$\overset{n<k<j}{=} \frac{c}{|c|}(-1+1) = 0 \qquad \forall n < k < j \tag{89b}$$

and then immediately see that

$$\partial_\ell m_{nj} = -\,|m_{nn} - m_{jj}| \cdot \underbrace{m_{nj}}_{=0}$$

$$+ \sum_{k \neq n, j} \underbrace{\left( \frac{m_{nn}^* - m_{kk}^*}{|m_{nn}^* - m_{kk}^*|} + \frac{m_{jj}^* - m_{kk}^*}{|m_{jj}^* - m_{kk}^*|} \right)}_{=0 \,\forall n<k<j} \cdot \underbrace{m_{nk} m_{kj}}_{=0 \,\forall k \notin [n,j]} \tag{90a}$$

$$= 0 \qquad \forall |n-j| > \delta \,, \tag{90b}$$

where the step $m_{nk} m_{kj} = 0 \,\forall k < n \vee k > j$ follows from the band-diagonality (88). $\square$

Note that (87) only states that while $M(\ell)$ is complex-sorted, the off-diagonals are not renormalized. If at some point the diagonal elements $m_{nn}(\ell)$ are not sorted anymore, which can happen, the band-diagonality is no longer preserved. Furthermore, most band-diagonal

matrices (88) are not complex-sorted, so that (89b) does not hold and the band-diagonality is not preserved, e.g for the matrix

$$M_1 = \begin{pmatrix} 0 & 1 & 0 \\ 1 & 1 & 1 \\ 0 & 1 & i \end{pmatrix}. \tag{91}$$

Actually, even for the Hermitian matrix

$$M_2 = \begin{pmatrix} 0 & 1 & 0 \\ 1 & 1 & 1 \\ 0 & 1 & 0 \end{pmatrix}, \tag{92}$$

the band-diagonality is not preserved by the gpc-generator, since the diagonal elements are not sorted in ascending order. We can achieve preservation of band-diagonality if we change the second and third basis vector

$$M_2' = \begin{pmatrix} 0 & 1 & 1 \\ 1 & 0 & 0 \\ 1 & 0 & 1 \end{pmatrix}, \tag{93}$$

in which case the band-diagonality of $M_2$ is meaningless since $M_2'$ has finite matrix elements on all minor diagonals. Note, that this sorting is only possible rigorously for matrices with real diagonal elements such as $M_2$, but not for general complex matrices such as $M_1$.

We showed that for a complex-sorted matrix $M(\ell)$ with diagonal width $\delta$, the components outside of its diagonal width are not renormalized $\partial_\ell m_{nj} = 0 \ \forall |n - j| > \delta$ for this specific value of $\ell$. For the band-diagonality to be preserved during the whole gpc-flow, that means for all $\ell$, the matrix must stay complex-sorted during the flow, i.e. $m_{nn}(\ell) = cx_n(\ell)$ with $x_{n+1}(\ell) > x_n(\ell)$. Special cases where this is true can be constructed, but it should be noted that in most cases, band-diagonality is no longer preserved.

In conclusion, the gpc-generator no longer preserves band-diagonality apart from some rare limiting cases.

## C Analytical Solution of the Single-Mode Model

In Sec. 4.2 we introduce a single-mode fermionic model with dissipation and discuss the stable fixed points of all four flows. An analytical solution is only discussed for the simplified case with $\Gamma_2 = 0$ in Sec. 4.2.3.

Here, we present a full analytical solution for arbitrary parameters. Additionally, we demonstrate how observables $O$ are transformed to $O_{\text{eff}}$ by performing the transformation analytically for a simple observable.

### C.1 Analytical Solution of the Lindbladian

We discuss an analytical solution of the flow equations

$$\partial_\ell \alpha = 2\mu_1 \mu_2 \text{sign}(\alpha), \tag{94a}$$
$$\partial_\ell \mu_1 = -2\mu_1 |\alpha|, \tag{94b}$$
$$\partial_\ell \mu_2 = -2\mu_2 |\alpha|, \tag{94c}$$

with the initial conditions

$$\alpha(0) = -\frac{\hbar(\Gamma_1 - \Gamma_2)}{2}, \tag{95a}$$

$$\mu_{1,2}(0) = \hbar\Gamma_{1,2}, \tag{95b}$$

that are discussed in Sec. 4.2. For brevity, we use the definitions

$$A := \frac{\hbar(\Gamma_1 + \Gamma_2)}{2}, \tag{96a}$$

$$B := -\frac{\Gamma_1 - \Gamma_2}{\Gamma_1 + \Gamma_2}. \tag{96b}$$

By using the second invariant of motion $\alpha^2 + \mu_1\mu_2 = A^2$ from (57) the flow equation for $\alpha$ can be formulated as

$$\partial_\ell \alpha = 2(A^2 - \alpha^2)\text{sign}(\alpha). \tag{97}$$

By separating the variables $\alpha$ and $\ell$ and using $\int (A^2 - \alpha^2)^{-1}\text{d}\alpha = \tanh^{-1}(\alpha/A)/A$ we find

$$\boxed{\alpha(\ell) = \text{sign}(\alpha(0))A\tanh\left[2A\ell + \text{sign}(\alpha(0))\tanh^{-1}(B)\right].} \tag{98}$$

Note that the sign of $\alpha$ does not change during the flow, i.e. $\text{sign}(\alpha(\ell)) = \text{sign}(\alpha(0)) \,\forall \ell$. The flow equations (94) and initial conditions (95) imply that $\mu_1(\ell)$ and $\mu_2(\ell)$ can be expressed by a single factor

$$\mu(\ell) := \mu_1(\ell) = \frac{\Gamma_1}{\Gamma_2}\mu_2(\ell). \tag{99a}$$

By inserting (98) in the second invariant of motion $\alpha^2 + \mu_1\mu_2 = A^2$ and applying trigonometric relations, we find

$$\boxed{\mu(\ell) = \sqrt{\frac{\Gamma_1}{\Gamma_2}}A \Big/ \cosh\left[2A\ell + \text{sign}(\alpha(0))\tanh^{-1}(B)\right].} \tag{100}$$

A quick check confirms that $\alpha(\infty) = \text{sign}(\alpha)A$ and $\mu(\infty) = 0$ are the stable fixed points discussed in Sec. 4.2.

## C.2 Basis Transformation of an Observable

The flow equation method does not only transform $M$, but instead performs a basis transformation. Therefore, all observables $O$ can be transformed to the new basis by using the same flow as $M$. As an instructive example, we perform an analytical transformation of the charge observable $\hat{O} = c^\dagger c$ with the matrix representation

$$O(0) = \begin{pmatrix} 1 & 0 \\ 0 & 0 \end{pmatrix} \quad \text{with } \hat{O}(\ell) = \begin{pmatrix} c^\dagger & \tilde{c} \end{pmatrix} O(\ell) \begin{pmatrix} c \\ \tilde{c}^\dagger \end{pmatrix}. \tag{101}$$

The parametrization

$$O(\ell) = \begin{pmatrix} \omega_1(\ell) & i\chi_2(\ell) \\ -i\chi_1(\ell) & \omega_2(\ell) \end{pmatrix} \quad \text{with } \omega_1(0) = 1, \ \omega_2(0) = \chi_1(0) = \chi_2(0) = 0, \tag{102}$$

yields closed flow equations $\partial_\ell O(\ell) = [\eta^{\text{gpc}}(\ell), O(\ell)]$ that read

$$\partial_\ell\omega_1 = -\partial_\ell\omega_2 = -2C\text{sign}(\alpha)\chi\mu, \qquad \text{with } C := \frac{\Gamma_2}{\Gamma_1}, \tag{103a}$$

$$\partial_\ell\chi = \text{sign}(\alpha)(\omega_1 - \omega_2)\mu, \quad \text{with } \chi := \chi_1 = \frac{\Gamma_1}{\Gamma_2}\chi_2. \tag{103b}$$

The flow equations and initial conditions imply $\omega_2(\ell) = 1 - \omega_1(\ell)$ and, therefore,

$$\partial_\ell \chi = \text{sign}(\alpha)(2\omega_1 - 1)\mu. \tag{104}$$

By solving $\partial_\ell \omega_1$ and $\partial_\ell \chi$ for $\mu$ and identifying the two terms with one another, we find

$$(2\omega_1 - 1)(\partial_\ell \omega_1) = -2C\chi(\partial_\ell \chi) \tag{105a}$$

$$\Leftrightarrow \omega_1^2 - \omega_1 = -C\chi^2 \tag{105b}$$

$$\Leftrightarrow \chi = \pm\sqrt{\frac{\omega_1(1-\omega_1)}{C}} \Leftrightarrow \omega_1 = \frac{1}{2} \pm \sqrt{\frac{1}{4} - C\chi^2}. \tag{105c}$$

For $\omega_1$, only the positive case

$$\omega_1 = \frac{1}{2} + \sqrt{\frac{1}{4} - C\chi^2}. \tag{106}$$

fulfills the initial conditions $\omega_1(0) = 1$ and $\chi(0) = 0$. With this and the analytical solution (100) of $\mu(\ell)$, the flow equation for $\chi$ can be reformulated as

$$\partial_\ell \chi = \sqrt{1 - 4C\chi^2}\mu \tag{107}$$

$$= \frac{A}{\sqrt{C}}\sqrt{1 - 4C\chi^2} \Big/ \cosh\left[2A\ell + \text{sign}(\alpha(0))\tanh^{-1}(B)\right], \tag{108}$$

$$\partial_{\ell'} \tilde{\chi}(\ell') = \sqrt{1 - \tilde{\chi}^2} \Big/ \cosh(\ell'), \tag{109}$$

with $\tilde{\chi} = 2\sqrt{C}\chi$ and $\ell' = 2A\ell + \text{sign}(\alpha(0))\tanh^{-1}(B)$. This expression can be further simplified by substituting $\tilde{\chi}(\ell') =: \sin(u(\ell'))$ and applying trigonometric relations. This flow equation is solved by

$$\tilde{\chi}(\ell') = \sin\left(2\tan^{-1}\tanh\left(\frac{\ell'}{2}\right) - B'\right), \tag{110a}$$

$$B' := 2\text{sign}(\alpha(0))\tan^{-1}\tanh\left(\frac{\tanh^{-1}(B)}{2}\right) = \text{sign}(\alpha(0))\sin^{-1}(B). \tag{110b}$$

The short expression $B' = \text{sign}(\alpha(0))\sin^{-1}(B)$ can be derived by applying the trigonometric relation $\sin(2\tan^{-1}(x)) = 2x/(x^2 + 1)$ to

$$\sin(B') = \sin\left[2\text{sign}(\alpha(0))\tan^{-1}\tanh\left(\frac{\tanh^{-1}(B)}{2}\right)\right] \tag{111a}$$

$$= \frac{2\tanh(\frac{1}{2}\tanh^{-1}(B))}{\tanh^2(\frac{1}{2}\tanh^{-1}(B)) + 1} \tag{111b}$$

$$= \tanh(2 \cdot \frac{1}{2}\tanh^{-1}(B)) \tag{111c}$$

$$= B, \tag{111d}$$

After reversing the transformations $\tilde{\chi} \to \chi$ and $\ell' \to \ell$, we obtain the solution

$$\chi(\ell) = \frac{1}{2\sqrt{C}}\sin\left[2\tan^{-1}\tanh\left(A\ell + \text{sign}(\alpha(0))\frac{1}{2}\tanh^{-1}(B)\right) - B'\right], \tag{112a}$$

$$\chi(\infty) = \frac{1}{2\sqrt{C}}\cos(B') = \frac{\sqrt{1 - B^2}}{2\sqrt{C}} = \text{sign}(\alpha)\frac{\Gamma_1}{\Gamma_1 + \Gamma_2}. \tag{112b}$$

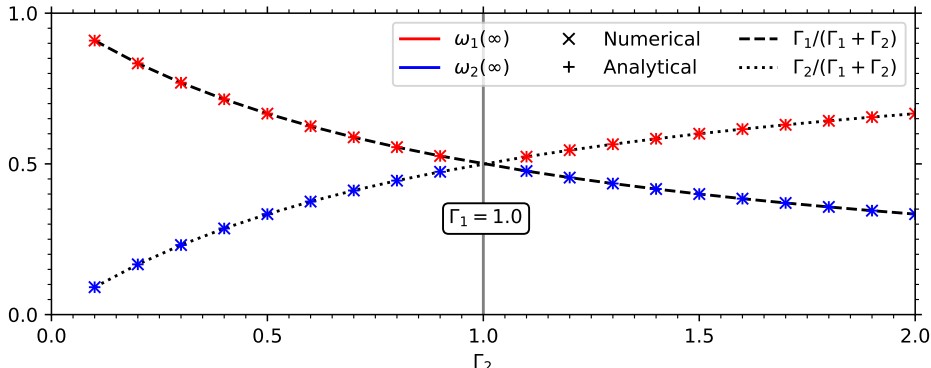

Figure 16: Comparison of the analytical results (113) with numerical results of $\omega_1(\infty)$ and $\omega_2(\infty)$ obtained by integrating the flow equations numerically. The data points match up to numerical precision and lie on the expected lines $\Gamma_1/(\Gamma_1+\Gamma_2)$ and $\Gamma_2/(\Gamma_1 + \Gamma_2)$.

By inserting (112b) in (106), we find

$$\omega_1(\infty) = \frac{1}{2} + \frac{1}{2}\sin(B') = \frac{1}{2}\left(1 + \frac{|\Gamma_1 - \Gamma_2|}{\Gamma_1 + \Gamma_2}\right), \tag{113a}$$

$$\omega_2(\infty) = \frac{1}{2} - \frac{1}{2}\sin(B') = \frac{1}{2}\left(1 - \frac{|\Gamma_1 - \Gamma_2|}{\Gamma_1 + \Gamma_2}\right). \tag{113b}$$

With this, we have derived the components of the effective observable

$$O_{\text{eff}} = \begin{pmatrix} \omega_1(\infty) & iC\chi(\infty) \\ -i\chi(\infty) & \omega_2(\infty) \end{pmatrix}, \tag{114a}$$

$$\hat{O}_{\text{eff}} = \omega_1(\infty)D^\dagger d + \omega_2(\infty)\tilde{D}\tilde{d}^\dagger + iC\chi(\infty)D^\dagger\tilde{d}^\dagger - i\chi(\infty)\tilde{D}d, \tag{114b}$$

with the operators $D^\dagger, \tilde{D}, d$ and $\tilde{d}^\dagger$ introduced in Sec. 4.2.

The charge of the steady state is $n_\infty := \langle I|c^\dagger c|\rho_\infty\rangle = \langle I|O|\rho_\infty\rangle$ in the original basis at $\ell = 0$. Ref. [1] discussed that the steady state $|\rho_\infty\rangle$ is defined by $d|\rho_\infty\rangle = \tilde{D}|\rho_\infty\rangle = 0$ in the effective basis at $\ell \to \infty$ and explains that the anticommutation relation $\{\tilde{D}, \tilde{d}^\dagger\} = 1$ can be applied to show $n_\infty = \omega_2(\infty)$. Note, however, that the flow orders the eigenvalues such that $\omega_2 < \omega_1$, so the basis depends on $\text{sign}(\alpha) = \text{sign}(\Gamma_2 - \Gamma_1)$. For $\Gamma_2 > \Gamma_1$ the basis states are switched and $n_\infty = \omega_1(\infty)$. Despite this, $\omega_1(\infty)$ and $\omega_2(\infty)$ match with the expected results $\Gamma_1/(\Gamma_1 + \Gamma_2)$ and $\Gamma_2/(\Gamma_1 + \Gamma_2)$, as can be seen in Fig. 16.

# D Analytical Properties of the Ordered Dissipative Scattering Model

We discuss analytical properties of the flow equations for the problem discussed in Sec. 4.4. To make use of symmetries, the matrix elements $m_{nj}$ are parametrized as

$$m_{nj}(\ell) = \epsilon_{nj}(\ell) + i\alpha_{nj}(\ell) \qquad \forall n, j \in [-N, -N+1, ..., N-1, N], \tag{115}$$

with $\epsilon_{nj}(\ell), \alpha_{nj}(\ell) \in \mathbb{R}$ and initial values

$$\epsilon_{nj}(0) = \delta_{nj}\hbar v \frac{2\pi}{L} n\,, \tag{116a}$$

$$\alpha_{nj}(0) = -\frac{\hbar^2 \gamma}{2L}\,. \tag{116b}$$

The flow equations (30b) read

$$\partial_\ell m_{nj} = -|m_{nn} - m_{jj}|m_{nj} + \sum_{k \neq n,j} \left( \frac{m_{nn}^* - m_{kk}^*}{|m_{nn}^* - m_{kk}^*|} + \frac{m_{jj}^* - m_{kk}^*}{|m_{jj}^* - m_{kk}^*|} \right) m_{nk} m_{kj}\,. \tag{117}$$

At $\ell = 0$ we observe

$$\boxed{\begin{array}{lll} m_{nj} = -m_{-n,-j}^* & \forall n, j \in [-N, -N+1, ..., N-1, N]\,, & \tag{118a} \\ m_{nj} = \phantom{-}m_{jn} & \forall n, j \in [-N, -N+1, ..., N-1, N]\,. & \tag{118b} \end{array}}$$

We claim that this is true $\forall \ell \in \mathbb{R}_0^+$ and prove these claims by showing that the derivatives $\partial_\ell m_{nj}$ fulfill the same symmetries, i.e.

$$\partial_\ell m_{nj} = -\partial_\ell m_{-n,-j}^*\,, \tag{119a}$$

$$\partial_\ell m_{nj} = \phantom{-}\partial_\ell m_{jn}\,. \tag{119b}$$

It suffices to prove (119) under the assumption of (118), since this implies that the flow equations (117) do not break the symmetries (119). First, we prove (119a). According to (117) we have

$$\partial_\ell m_{-n,-j} = -|m_{-n,-n} - m_{-j,-j}|m_{-n,-j}$$
$$+ \sum_{k \neq n,j} \left( \frac{m_{-n,-n}^* - m_{-k,-k}^*}{|m_{-n,-n}^* - m_{-k,-k}^*|} + \frac{m_{-j,-j}^* - m_{-k,-k}^*}{|m_{-j,-j}^* - m_{-k,-k}^*|} \right) m_{-n,-k} m_{-k,-j} \tag{120a}$$

$$\overset{(118a)}{=} |m_{nn} - m_{jj}|m_{nj}^* - \sum_{k \neq n,j} \left( \frac{m_{nn} - m_{kk}}{|m_{nn}^* - m_{kk}^*|} + \frac{m_{jj} - m_{kk}}{|m_{jj}^* - m_{kk}^*|} \right) m_{nk}^* m_{kj}^* \tag{120b}$$

$$= -\partial_\ell m_{n,j}^*\,. \tag{120c}$$

Note that we applied $|z_1 - z_2| = |z_1^* - z_2^*| \; \forall z_i \in \mathbb{C}$ in line (120b). To prove (119b), we apply (117)

$$\partial_\ell m_{jn} = -|m_{nn} - m_{jj}|m_{jn} + \sum_{k \neq n,j} \left( \frac{m_{nn}^* - m_{kk}^*}{|m_{nn}^* - m_{kk}^*|} + \frac{m_{jj}^* - m_{kk}^*}{|m_{jj}^* - m_{kk}^*|} \right) m_{jk} m_{kn} \tag{121a}$$

$$\overset{(118b)}{=} -|m_{nn} - m_{jj}|m_{nj} + \sum_{k \neq n,j} \left( \frac{m_{nn}^* - m_{kk}^*}{|m_{nn}^* - m_{kk}^*|} + \frac{m_{jj}^* - m_{kk}^*}{|m_{jj}^* - m_{kk}^*|} \right) m_{kj} m_{nk} \tag{121b}$$

$$= \partial_\ell m_{nj}\,. \tag{121c}$$

With this, we showed that the flow equations do not break the symmetries (118). Therefore, the symmetries are fulfilled $\forall \ell \in \mathbb{R}_0^+$.

The symmetries (118) imply that all eigenvalues appear in pairs $(\lambda, -\lambda^*)$, aside from the singular eigenvalue $\lambda_0 = i\alpha_{00}(\infty)$ with $\epsilon_{00}(\ell) = 0$. By applying (117), the general derivatives

of $\epsilon_{nk}$ and $\alpha_{nk}$ read

$$\partial_\ell \epsilon_{nj} = -|m_{nn} - m_{jj}|\epsilon_{nj} + \mathrm{Re}\left[\sum_{k\neq n,j}\left(\frac{m_{nn}^* - m_{kk}^*}{|m_{nn}^* - m_{kk}^*|} + \frac{m_{jj}^* - m_{kk}^*}{|m_{jj}^* - m_{kk}^*|}\right)m_{nk}m_{kj}\right], \tag{122a}$$

$$\partial_\ell \alpha_{nj} = -|m_{nn} - m_{jj}|\alpha_{nj} + \mathrm{Im}\left[\sum_{k\neq n,j}\left(\frac{m_{nn}^* - m_{kk}^*}{|m_{nn}^* - m_{kk}^*|} + \frac{m_{jj}^* - m_{kk}^*}{|m_{jj}^* - m_{kk}^*|}\right)m_{nk}m_{kj}\right]. \tag{122b}$$

The derivative of $\alpha_{00}(\ell)$ reads

$$\partial_\ell \alpha_{00} = 2\,\mathrm{Im}\left[\sum_{k\neq 0}\frac{m_{00}^* - m_{kk}^*}{|m_{00}^* - m_{kk}^*|}m_{0k}^2\right] \tag{123a}$$

$$= 4\,\mathrm{Im}\left[\sum_{k>0}\frac{-\epsilon_{kk} + \mathrm{i}(\alpha_{kk} - \alpha_{00})}{\sqrt{\epsilon_{kk}^2 + (\alpha_{00} - \alpha_{kk})^2}}(\epsilon_{0k} + \mathrm{i}\alpha_{0k})^2\right] \tag{123b}$$

$$= 4\sum_{k>0}\frac{-2\epsilon_{kk}\epsilon_{0k}\alpha_{0k} + (\alpha_{kk} - \alpha_{00})(\epsilon_{0k}^2 - \alpha_{0k}^2)}{\sqrt{\epsilon_{kk}^2 + (\alpha_{00} - \alpha_{kk})^2}}. \tag{123c}$$

Further simplifications do not seem possible without knowledge about the solutions $m_{0k}(\ell)$ and $m_{kk}(\ell)$.

# E   Automated Flow Equation Calculation for the Ordered Dissipative Scattering Model

We show here that setting up the flow equations for the third example in Sec. 4.4 by commuting the matrices analytically leads to exactly the same flow equations as the second quantization formulation in Ref. [1], as expected. The R2-generator for the problem reads

$$\eta_{ab}^{\mathrm{R2}} = [D^\dagger, V]_{ab} = \sum_c d_{ac}^* v_{cb} - v_{ac}d_{cb}^*, \tag{124}$$

and the flow

$$\partial_\ell m_{kq}^{\mathrm{R2}} = [\eta^{\mathrm{R2}}, M]_{kq} \tag{125a}$$

$$= \sum_{s,c}(d_{kc}^* v_{cs} - v_{kc}d_{cs}^*)(d_{sq} + v_{sq}) - (d_{ks} + v_{ks})(d_{sc}^* v_{cq} - v_{sc}d_{cq}^*) \tag{125b}$$

$$= \sum_s(-v_{ks}d_{ss}^*)(d_{sq} + v_{sq}) - (d_{ks} + v_{ks})(d_{ss}^* v_{sq})$$
$$+ \sum_{s,c\neq s}(d_{kc}^* v_{cs})(d_{sq} + v_{sq}) - (d_{ks} + v_{ks})(-v_{sc}d_{cq}^*) \tag{125c}$$

$$= (-v_{kq}d_{qq}^*)d_{qq} - d_{kk}(d_k^* v_{kq}) + \sum_{s\neq q}(-v_{ks}d_{ss}^*)v_{sq} - \sum_{s\neq k}v_{ks}(d_{ss}^* v_{sq})$$
$$+ \sum_{c\neq q}(d_{kc}^* v_{cq})d_{qq} + \sum_{c\neq k}d_{kk}(v_{kc}d_{cq}^*) + \sum_{s\neq q,c\neq s}(d_{kc}^* v_{cs})v_{sq} + \sum_{s\neq k,c\neq s}v_{ks}(v_{sc}d_{cq}^*) \tag{125d}$$

$$= -v_{kq}(|d_{qq}|^2 + |d_{kk}|^2 - d_{kk}^* d_{qq} - d_{kk}d_{qq}^*) + \sum_{s\neq\{k,q\}}v_{ks}v_{sq}(d_{kk}^* + d_{qq}^* - 2d_{ss}^*) \tag{125e}$$

$$= -v_{kq}|d_{qq} - d_{kk}|^2 + \sum_{s\neq\{k,q\}}v_{ks}v_{sq}(d_{kk}^* + d_{qq}^* - 2d_{ss}^*), \tag{125f}$$

$$\Rightarrow \partial_\ell m_{kk}^{\mathrm{R2}} = 2\sum_{s\neq k}v_{ks}v_{sk}(d_{kk}^* - d_{ss}^*), \tag{125g}$$

is identical to the flow calculated using second quantization in Ref. [1], except for a factor i, which was included in the non-diagonal components in Ref. [1]. The missing restriction $s \neq \{k, q\}$ in Ref. [1] is a typo, which was confirmed by the authors.

The R3-generator reads

$$\eta_{ab}^{\mathrm{R3}} = \begin{cases} v_{ab}/(d_{aa} - d_{bb}), & \text{for } d_{aa} \neq d_{bb}, \\ 0, & \text{for } d_{aa} = d_{bb}, \end{cases} \tag{126}$$

and the corresponding flow is given by

$$\partial_\ell m_{kq}^{\mathrm{R3}} = [\eta^{\mathrm{R3}}, M]_{kq} \tag{127a}$$

$$= \sum_s \left( \eta_{ks}(d_{sq} + v_{sq}) - (d_{ks} + v_{ks})\eta_{sq} \right) \tag{127b}$$

$$= \sum_{s \neq k} \frac{v_{ks}(d_{sq} + v_{sq})}{d_{kk} - d_{ss}} - \sum_{s \neq q} \frac{(d_{ks} + v_{ks})v_{sq}}{d_{ss} - d_{qq}} \tag{127c}$$

$$= \frac{v_{kq}d_{qq}}{d_{kk} - d_{qq}} - \frac{d_{kk}v_{kq}}{d_{kk} - d_{qq}} + \sum_{s \neq \{k,q\}} \left( \frac{v_{ks}v_{sq}}{d_{kk} - d_{ss}} + \frac{v_{ks}v_{sq}}{d_{qq} - d_{ss}} \right) \tag{127d}$$

$$= -v_{kq} + \sum_{s \neq \{k,q\}} \left( v_{ks}v_{sq} \frac{d_{kk} + d_{qq} - 2d_{ss}}{(d_{kk} - d_{ss})(d_{qq} - d_{ss})} \right) \tag{127e}$$

$$\partial_\ell m_{kk}^{\mathrm{R3}} = 2 \sum_{s \neq k} \frac{v_{ks}v_{sk}}{d_{kk} - d_{ss}}, \tag{127f}$$

which again matches the result in second quantization [1]. We stress that this shows that calculating the flow equations using automated numerical matrix multiplications without changing to second quantization is a feasible approach that does not neglect any relevant physics. In real applications beyond the scope of the benchmarks presented here, however, flow equations are commonly calculated and solved in second quantization, because this approach allows one to tread more processed with less parameters.

# F Sampling of Random Lindbladians

In Sec. 4.6 we introduce

$$\mathcal{L}_D(\rho) = \sum_{m,n=1}^{N^2-1} K_{mn} \left[ F_n \rho F_m^\dagger - \frac{1}{2} \left( F_m^\dagger F_n \rho + \rho F_m^\dagger F_n \right) \right] \tag{128}$$

as a way to sample random Lindbladians. Here, we explain the method in more detail.

The Hilbert-Schmidt basis $\{F_n\}, n = 1, 2, ..., N^2 - 1$ is traceless ($\mathrm{Tr}(F_n) = 0$) and orthonormal ($\mathrm{Tr}(F_n F_m^\dagger) = \delta_{n,m}$). We can sample them using the $N^2 - 1$ Hermitian generators of SU($N$)

$$\frac{N(N-1)}{2} \text{ symmetric matrices} \quad S_{jk} = \frac{1}{\sqrt{2}} \left( |j\rangle\langle k| + |k\rangle\langle j| \right), \tag{129a}$$

$$\frac{N(N-1)}{2} \text{ antisymmetric matrices} \quad J_{jk} = -\frac{\mathrm{i}}{\sqrt{2}} \left( |j\rangle\langle k| - |k\rangle\langle j| \right), \tag{129b}$$

$$N - 1 \text{ diagonal matrices} \quad D_l = \frac{1}{\sqrt{l(l+1)}} \left( \sum_{n=1}^{l} |n\rangle\langle n| - l|l+1\rangle\langle l+1| \right), \tag{129c}$$

$$\text{with } j, k \in \{1, 2, ..., N\} \text{ and } l \in \{1, 2, ..., N-1\}. \tag{129d}$$

In $N = 2$ these are the well-known Pauli matrices and in $N = 3$ the standard eight Gell-Mann matrices.

All matrices $K$, $F_n$ and $\rho$ are of dimension $N \times N$. After constructing $\mathcal{L}_D(\rho)$, we calculate all its matrix elements by using a basis $\{B_n\}_n$ for all possible density matrices $\rho$ in Fock-Liouville space. This basis does not need to fulfill any particular conditions, so we simply choose $N^2$ possible matrices that have one element 1 and all other elements 0. After calculating the $(N^2 \times N^2)$-dimensional supermatrix representation of $\mathcal{L}_D(\rho)$, we diagonalize the supermatrix to receive the spectrum.

The computationally most costly part is calculating the matrix representation of $\mathcal{L}_D(\rho)$ which scales according to $\mathcal{O}(N^7)$ in the worst case, assuming an $\mathcal{O}(N^3)$ complexity for matrix multiplications. Considering that $F_n$ and $\rho$ are rather sparse and matrix products are often calculated more efficiently for sparse matrices, the real scaling is possibly slightly lower on average, i.e. $\mathcal{O}(N^\alpha)$ with non-integer scaling dimension $\alpha \in [6, 7]$. We optimized the sampling of the Lindbladians by using sparse matrix representations and pre-calculating all possible products $F_n \rho$ and $\rho F_m^\dagger$ outside of the innermost loop which sums over all possible values of $n$ and $m$ and calculates the matrix products. The matrix products are calculated using the Eigen library for C++ [69]. With these optimizations, the time necessary for sampling the Lindbladians was negligible compared to the time for solving the flow equations.

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
