# Peer review of "Efficient flow equations for dissipative systems"

_SciPost Physics, doi:SciPost Phys. 13, 122 (2022)_

## Round 1 · Referee Report · Anonymous (Referee 1) · 2022-7-4

Strengths

1) Well written manuscript, well presented numerical analysis 2)Detailed comparison of different generators, emphasizing strength and weaknesses 3)Frame the new GPC generator within a family of generators that includes previously introduced ones.

Weaknesses

1) Lacks a clear example where the proposed generator captures some physics which is missed by the previously introduced ones. 2)Not clear whether the proposed generator can be handled analytically (see report). 3)Literature on flow-equation applications to nonequilibrium problems should be improved.

Report

The manuscript by Ferkinghoff and Uhrig deals with open quantum systems described by Linbdlad master equations (or more generally non-unitary operators). The main focus of this work is to extend the dissipative flow equation approach introduced in Ref [1], by means of a new type of generator for the flow (generalised particle conserving generator, GPC).

The authors motivate the use of this generator, introduced for the study of ground-state problems and extended here for non-Hermitian matrices. The authors compare the GPC performances to the generators used in Ref.[1], providing evidence that their GPC generator offers a trade off between convergence time and truncation error.

I find this work interesting and potentially useful. In my opinion however it lacks new physics results that show the advantage of this new generator.
It is also not clear the general scope and possible applications of this generator. For example, the use of Wegner generator has allowed analytical many-body (second quantization) applications of flow-equation methods in the thermodynamic limit, based on physically motivated parametrization of the generator. Can the same be done for the GPC one? Or it will be limited to numerical diagonalization of non-hermitian (or Lindblad) operators for finite size matrices?

I certainly find one or more new research results which advance our current knowledge on a difficult and challenging open problem.
However I do not see the content of the manuscript as a groundbreaking theoretical or computational discovery, nor I am convinced that the newly proposed generator can qualify as a breakthrough result in the field.
Based on the points above I cannot recommend this work for SciPost Physics but it can be suitable for SciPost Physics Core.

Requested changes

1)Clarify the definition of PC generator in Section 2.2. Operator Q is not really defined when introduced in Eq 3a,b. Nor H_plus and H_minus. Given the importance of this generator for the rest of the paper, it would be useful to clarify its definition

2)Comment on the possible applications of the GPC generator to other dissipative models. Comment on whether the GPC generator allows for analytical (second quantization) approaches to dissipative flow-equation.

3)Improve the literature on flow equation applications given in the introduction, which is currently lacking several recent and not so recent works. Missing references include:

Flow equation for unitary dynamics such as
S. Kehrein, Scaling and decoherence in the nonequilibrium kondo model, Phys. Rev. Lett. 95, 056602 (2005)
M. Moeckel and S. Kehrein, Interaction quench in the hubbard model, Phys. Rev. Lett.100, 175702 (2008)

Flow Equation for Floquet systems (in addition to Ref 49-50)
SJ Thomson, D. Magano, M. Schiro SciPost Phys. 11, 028 (2021)

Flow Equation for MBL systems (in addition to Ref 56):
C. Monthus, Flow towards diagonalization for many-body-localization models: adaptation of the toda matrix differential flow to random quantum spin chains, Journal of Physics A: Mathematical and Theoretical 49(30), 305002 (2016),
D. Pekker, B. K. Clark, V. Oganesyan and G. Refael, Fixed points of wegner-
wilson flows and many-body localization, Phys. Rev. Lett. 119, 075701 (2017)
S. J. Thomson and M. Schiro, Time evolution of many-body localized systems with the flow equation approach, Phys. Rev. B 97, 060201 (2018)

  • validity: high
  • significance: good
  • originality: good
  • clarity: high
  • formatting: perfect
  • grammar: perfect

Author:  Gary Ferkinghoff  on 2022-07-26  [id 2686]

(in reply to Report 1 on 2022-07-04)

Reply to Referee 1

We thank the Referee for carefully reading our manuscript and the detailed report which helps us to improve the manuscript further.

Strengths 1) Well written manuscript, well presented numerical analysis 2)Detailed comparison of different generators, emphasizing strength and weaknesses 3)Frame the new GPC generator within a family of generators that includes previously introduced ones.

We are glad and grateful that the Referee finds our manuscript well written and providing a detailed comparison of the different generators and the numerical results.

Weaknesses 1) Lacks a clear example where the proposed generator captures some physics which is missed by the previously introduced ones. 2)Not clear whether the proposed generator can be handled analytically (see report). 3)Literature on flow-equation applications to nonequilibrium problems should be improved.

We thank the Referee for the constructive criticism and address further points below.

Report The manuscript by Ferkinghoff and Uhrig deals with open quantum systems described by Linbdlad master equations (or more generally non-unitary operators). The main focus of this work is to extend the dissipative flow equation approach introduced in Ref [1], by means of a new type of generator for the flow (generalised particle conserving generator, GPC). The authors motivate the use of this generator, introduced for the study of ground-state problems and extended here for non-Hermitian matrices. The authors compare the GPC performances to the generators used in Ref. [1], providing evidence that their GPC generator offers a trade off between convergence time and truncation error. I find this work interesting and potentially useful. In my opinion however it lacks new physics results that show the advantage of this new generator.

We are glad that the Referee assesses our work to be interesting. Concerning the point of “new physics”, we agree that this is an excellent add-on. We point out that our study provides interesting insights in the renormalization flow of generic Lindbladians which differ from the one of generic non-Hermitian Hamiltonians. But clearly the focus of the present manuscript is to discuss the crucial properties of various generator schemes and we agree with the Referee that this field is still in its very infancy. In this respect, we emphasize that the introduction of a novel generator scheme is only one of two key points of our work. The second, equally relevant point, is to discuss the robustness against truncation linked to the renormalizing property of flow equations. This was no topic in the work by Rosso et al. so that we feel that our contribution fills a vital gap in the study of flow equations for dissipative systems. Thereby, a new pathway in a currently intensively pursued direction of research is opened with clear potential for a wide range of follow-up work. Thus, we think that SciPost Physics is a perfectly appropriate journal for publication of our manuscript.

It is also not clear the general scope and possible applications of this generator. For example, the use of Wegner generator has allowed analytical many-body (second quantization) applications of flow-equation methods in the thermodynamic limit, based on physically motivated parametrization of the generator. Can the same be done for the GPC one? Or it will be limited to numerical diagonalization of non-hermitian (or Lindblad) operators for finite size matrices?

We understand that the Referee is concerned about the analytical applicability of the proposed generator. In this respect, we point out that the examples in Secs. 4.2, 4.4, and 4.5 are formulated in second quantization and the superoperator formalism which ensures that the physics of very high-dimensional Hilbert spaces is captured. In order to stress this aspect, we added explanatory remarks in Secs. 4.2.1 and 4.4.1. It is true that the actual calculations for the flow of the matrices representing the prefactors of terms in second quantization were performed numerically. In order to illustrate that analytical arguments are also possible we added Appendix C with the analytical solution of the model from Sec. 4.2 including the transformation of an observable and Appendix D establishing analytical properties of the model of Sec. 4.4.

I certainly find one or more new research results which advance our current knowledge on a difficult and challenging open problem. However I do not see the content of the manuscript as a groundbreaking theoretical or computational discovery, nor I am convinced that the newly proposed generator can qualify as a breakthrough result in the field. Based on the points above I cannot recommend this work for SciPost Physics but it can be suitable for SciPost Physics Core.

We respectfully disagree with the Referee on this point. We feel that our study contributes significantly to the emergent field of flow equations for dissipative systems, thereby “opening a new pathway in an existing or a new research direction, with clear potential for multipronged follow-up work” which warrants publication in SciPost Physics. The two key achievements are the introduction of a novel generator and (to our knowledge) the first comprehensive consideration of robustness against truncation errors in the field of dissipative systems. We consider both points to be equally important.

Requested changes 1) Clarify the definition of PC generator in Section 2.2. Operator Q is not really defined when introduced in Eq 3a,b. Nor H_plus and H_minus. Given the importance of this generator for the rest of the paper, it would be useful to clarify its definition

The text below Equations 3a, b (which are labelled 4a, b in the revised manuscript) discusses that various definitions of the operator Q are valid and provides generic examples. But we agree that this explanation was given only after discussing other details of the operator Q. This may confuse readers expecting a more concrete definition. In the revised manuscript, we added an explanation closer to Equations 4a, b which clarifies that the concrete definition of Q is chosen according to the problem under study. The operators H+ and H- were defined only verbally so far. We modified this by adding Equation 5 defining the operators more strictly. A sentence explaining how H+ and H- are defined if Q counts the number of quasi-particles is also appended.

2)Comment on the possible applications of the GPC generator to other dissipative models. Comment on whether the GPC generator allows for analytical (second quantization) approaches to dissipative flow-equation.

As discussed above, the models in Secs. 4.2, 4.4, and 4.5 are formulated in second quantization. Furthermore, the revised manuscript addresses analytical results in Appendix C and Appendix D.

3)Improve the literature on flow equation applications given in the introduction, which is currently lacking several recent and not so recent works. Missing references include:

Flow equation for unitary dynamics such as S. Kehrein, Scaling and decoherence in the nonequilibrium kondo model, Phys. Rev. Lett. 95, 056602 (2005) M. Moeckel and S. Kehrein, Interaction quench in the hubbard model, Phys. Rev. Lett.100, 175702 (2008)

Flow Equation for Floquet systems (in addition to Ref 49-50) SJ Thomson, D. Magano, M. Schiro SciPost Phys. 11, 028 (2021)

Flow Equation for MBL systems (in addition to Ref 56): C. Monthus, Flow towards diagonalization for many-body-localization models: adaptation of the toda matrix differential flow to random quantum spin chains, Journal of Physics A: Mathematical and Theoretical 49(30), 305002 (2016), D. Pekker, B. K. Clark, V. Oganesyan and G. Refael, Fixed points of wegner- wilson flows and many-body localization, Phys. Rev. Lett. 119, 075701 (2017) S. J. Thomson and M. Schiro, Time evolution of many-body localized systems with the flow equation approach, Phys. Rev. B 97, 060201 (2018)

We thank the Referee for pointing out additional references that are very useful for the topics of the paper. We included them in the Introduction.

---

## Round 1 · Referee Report · Anonymous (Referee 2) · 2022-7-7

Report

This paper discusses a generalization of the Wegner-Wilson-Glazek unitary flow equation method to non-Hermitean matrices. Non-Hermitean matrices appear naturally as the Lindbladians describing the time evolution of open quantum systems. Naively using the Wegner generator for the unitary flow does not work since the flow will generally not converge to the diagonal basis. Suitable generalizations for the Wegner generator were first introduced by Rosso et al. in 2020. The present paper presents another choice for the generator, which the authors call generalized particle-conserving generator (gpc-generator). The whole topic is still in its infancy as the authors correctly point out, and the present paper presents a comprehensive comparative study of these different generators suggested for non-Hermitean matrices. There is a dichotomy between stability against (truncation) errors and convergence speed, and this is tested extensively for various models.

The paper is clearly written and shows very good understanding of the flow equation methodology. It will be a very good resource for readers who want to apply this machinery to the many important problems in open quantum systems. There is only one criticism that I would like to make. In the flow equation methodology it is important to keep in mind that one is changing the basis along the flow, therefore also transforming the observables. It is important to emphasize this point and I strongly recommend to add explanatory remarks along these lines. What would be very nice (but not required) is to study this explicitly for the model in chapter 4.2 (fermionic mode with losses and gains), especially since such transformations of observables have so far only been studied for unitary (vs. similarity) transformations.

I recommend that the paper should be published once the above comment regarding the transformation of observables is included.
  • validity: -
  • significance: -
  • originality: -
  • clarity: -
  • formatting: -
  • grammar: -

Author:  Gary Ferkinghoff  on 2022-07-26  [id 2687]

(in reply to Report 2 on 2022-07-07)

Reply to Referee 2

We thank the Referee for reading our manuscript and providing valuable feedback.

Report This paper discusses a generalization of the Wegner-Wilson-Glazek unitary flow equation method to non-Hermitean matrices. Non-Hermitean matrices appear naturally as the Lindbladians describing the time evolution of open quantum systems. Naively using the Wegner generator for the unitary flow does not work since the flow will generally not converge to the diagonal basis. Suitable generalizations for the Wegner generator were first introduced by Rosso et al. in 2020. The present paper presents another choice for the generator, which the authors call generalized particle-conserving generator (gpc-generator). The whole topic is still in its infancy as the authors correctly point out, and the present paper presents a comprehensive comparative study of these different generators suggested for non-Hermitean matrices. There is a dichotomy between stability against (truncation) errors and convergence speed, and this is tested extensively for various models. The paper is clearly written and shows very good understanding of the flow equation methodology. It will be a very good resource for readers who want to apply this machinery to the many important problems in open quantum systems.

We are very glad that the Referee judges our manuscript to be a valuable resource for its readers and all colleagues interested in open quantum systems.

There is only one criticism that I would like to make. In the flow equation methodology it is important to keep in mind that one is changing the basis along the flow, therefore also transforming the observables. It is important to emphasize this point and I strongly recommend to add explanatory remarks along these lines.

We thank the Referee for pointing this out and agree that the manuscript did not clarify that observables are transformed as well, which could confuse readers that are not familiar with the method. We added explanations and the equation for flow equations of observables in Sec. 2.1 and Sec. 2.3 .

What would be very nice (but not required) is to study this explicitly for the model in chapter 4.2 (fermionic mode with losses and gains), especially since such transformations of observables have so far only been studied for unitary (vs. similarity) transformations.

We agree that the model in Sec. 4.2 provides a nice example to study the transformation of an observable explicitly. We added a new appendix, labelled App. C in the revised manuscript, in which we discuss an analytical solution of the flow equations for the Lindbladian and the charge observable in Sec. 4.2 .

I recommend that the paper should be published once the above comment regarding the transformation of observables is included.

We are glad about the positive recommendation of the Referee. We addressed the comment regarding the transformation of observables and modified the manuscript accordingly.

---

## Round 2 · Referee Report · Anonymous (Referee 1) · 2022-9-7

Report

I have read the authors reply and their updated manuscript.

The authors have addressed the majority of the points that I raised. In particular the authors have added new material and analytical insights that could help developing the dissipative flow equation as a semi-analytical approach - rather than a brute force numerical method. The presentation has been also improved and so the review of the literature.

Based on the Authors reply and new material I am convinced that the criteria for publication in SciPost Physics have been met, so I am happy to recommend the paper for publication.

---

## Round 2 · Author Response

Dear Editor,

first of all, we like to point out that the first author has changed his name following his marriage. He would like to change his name on the manuscript from “Gary Ferkinghoff” to “Gary Schmiedinghoff”.
We thank you for taking up the time-consuming burden to provide two meaningful reports. We read both of them closely and modified our manuscript according to all the uttered constructive criticisms. In this way, the manuscript has improved and we think that it is now ready for publication.
Referee 1 sees a number of strengths (“well written manuscript”, “detailed compa¬rison”, “new GPC generator”). But she/he identified also three weaknesses (“no new physics”, “analytical handling”, “literature”) and missed a “groundbreaking theoretical and computational discovery” so that the recommendation is rather for SciPost Physics Core than for SciPost Physics.
We added the requested literature and point out that most of the studied examples already treat systems in second quantization already. We appended also analytical solutions where available so that two of three weaknesses are removed. As for new physics, we point out that our results provide novel insights in the renormalization of Lindbladians which partly differ significantly from the renormalization of non-Hermitian Hamiltonians. But we stress that the main aim of our study is to pave the way for new theoretical tools which have to be bench-marked with known results. Further¬more, we did not only introduce a novel generator scheme, but provided the first compre¬hen¬sive comparison of generators for open quantum systems with respect to their truncation properties; for instance, no such discussion is present in Rosso et al. (Ref. 1). This is the second key point of our manuscript. Its novelty in the field of dissipative quantum systems may have been underestimated by Referee 1.
Referee 2 is extremely positive about the manuscript: “The paper is clearly written and shows very good understanding of the flow equation methodology. It will be a very good resource for readers who want to apply this machinery to the many important problems in open quantum systems.” Hence, except for an amendment which we now have introduced in the revised manuscript, she/he “recommend[s] that the paper should be published once the above comment regarding the transformation of observables is included”.
All in all, we feel that our contribution provides a new approach in the recently booming field of quantum dissipation and non-equilibrium physics which clearly has the potential for a humongous variety of follow-up work. For this reason, we feel that our manuscript fits nicely to SciPost Physics.
We are looking forward to the advancing review process.

Yours sincerely
Gary Schmiedinghoff and Götz Uhrig

---

## Round 2 · List of Changes

1) Author name "Gary Ferkinghoff" was changed to "Gary Schmiedinghoff" 2) References 41, 42, 53, 60, 61 and 62 were added in the introduction 3a) In Sec. 2.1 a sentence was added in the fourth line, which points out that all relevant observables are transformed to effective observables in the same basis as the effective Hamiltonian 3b) Equation (3) and a short sentence in front of Eq. (3) were added 4a) In Sec. 2.2 a sentence was added two lines after Eq. (4), which points out that the concrete definition of Q is specific to the problem 4b) Equation (5) and an introductory sentence were added 4c) The text between Eq. (5) and Eq. (6) was slightly edited and a sentence explaining H+ and H- in more detail were added 5) In Sec 2.3 equation (13) was extended with the definition of the observable flow 6a) At the end of Sec. 4.2.1 a sentence stressing that the calculation is performed in second quantization was added 6b) Appendix C was added, which covers the analytical solution of the model in Sec. 4.2 and the analytical solution of an additional observable 6c) At the end of Sec. 4.2.3 the new appendix C is referenced 7a) In Sec. 4.4.1 after Eq. (67) a sentence stressing that the calculation is performed in second quantization was added 7b) Appendix D was added, which covers analytical properties of the flow in Sec. 4.4 7c) In front of Eq. (69) the new appendix D is referenced

---

## Editorial Decision

published